# PATL2 is a key actor of oocyte maturation whose invalidation causes infertility in women and mice

Marie Christou-Kent[1], Zine-Eddine Kherraf[1], Amir Amiri-Yekta[1,2,3], Emilie Le Blévec[1], Thomas Karaouzène[1], Béatrice Conne[4], Jessica Escoffier[1], Said Assou[5], Audrey Guttin[6], Emeline Lambert[1], Guillaume Martinez[1,2,7], Magalie Boguenet[1], Selima Fourati Ben Mustapha[8], Isabelle Cedrin Durnerin[9], Lazhar Halouani[8], Ouafi Marrakchi[8], Mounir Makni[8], Habib Latrous[8], Mahmoud Kharouf[8], Charles Coutton[1,2,7], Nicolas Thierry-Mieg[10], Serge Nef[4], Serge P Bottari[1], Raoudha Zouari[8], Jean Paul Issartel[6], Pierre F Ray[1,2,†] (ID) & Christophe Arnoult[1,*,†] (ID)

## Abstract

The genetic causes of oocyte meiotic deficiency (OMD), a form of primary infertility characterised by the production of immature oocytes, remain largely unexplored. Using whole exome sequencing, we found that 26% of a cohort of 23 subjects with OMD harboured the same homozygous nonsense pathogenic mutation in *PATL2*, a gene encoding a putative RNA-binding protein. Using *Patl2* knockout mice, we confirmed that PATL2 deficiency disturbs oocyte maturation, since oocytes and zygotes exhibit morphological and developmental defects, respectively. PATL2's amphibian orthologue is involved in the regulation of oocyte mRNA as a partner of CPEB. However, Patl2's expression profile throughout oocyte development in mice, alongside colocalisation experiments with Cpeb1, Msy2 and Ddx6 (three oocyte RNA regulators) suggest an original role for Patl2 in mammals. Accordingly, transcriptomic analysis of oocytes from WT and *Patl2*⁻/⁻ animals demonstrated that in the absence of Patl2, expression levels of a select number of highly relevant genes involved in oocyte maturation and early embryonic development are deregulated. In conclusion, PATL2 is a novel actor of mammalian oocyte maturation whose invalidation causes OMD in humans.

**Keywords** female sterility; oocyte developmental competence; oocyte maturation arrest; oocyte maturation failure; Patl2
**Subject Categories** Genetics, Gene Therapy & Genetic Disease; Urogenital System

## Introduction

In humans, oocyte production is a lengthy process that begins during embryonic development and is characterised by a long diapause lasting over a decade until resumption of maturation at puberty. The quiescent oocytes, contained within primordial follicles, are arrested in the prophase of meiosis I. Periodically, a group of primordial follicles are recruited to the pool of growing follicles. The germinal vesicle (GV) oocyte and surrounding follicular cells develop in tight coordination to produce a fully grown GV oocyte within an antral follicle. This process takes around 290 days Williams & Erickson, 2012; Li & Albertini, 2013). At this stage, the oocyte is sensitive to hormonal stimulation, which causes meiosis to resume, as revealed by GV breakdown (GVBD), and extrusion of the first polar body before arresting again at the metaphase 2 (MII) stage of meiosis II. The second meiosis is completed, with exclusion of the second polar body, upon fertilisation.

Several reports have been published describing cases of infertile women whose ovaries repeatedly produce mostly/only immature oocytes. This poorly defined syndrome is known as "oocyte factor infertility" or "bad eggs syndrome" (Hartshorne *et al*, 1999; Levran

1 Genetics, Epigenetics and Therapies of Infertility, Institute for Advanced Biosciences, Inserm U1209, CNRS UMR 5309, Université Grenoble Alpes, Grenoble, France
2 UM GI-DPI, CHU de Grenoble, Grenoble, France
3 Department of Genetics, Reproductive Biomedicine Research Center, Royan Institute for Reproductive Biomedicine, ACECR, Tehran, Iran
4 Department of Genetic Medicine and Development, University of Geneva Medical School, Geneva, Switzerland
5 IRMB, INSERM U1183, CHRU Montpellier, Université Montpellier, Montpellier, France
6 Grenoble Neuroscience Institute, INSERM 1216, Université Grenoble Alpes, Grenoble, France
7 UM de Génétique Chromosomique, CHU de Grenoble, Grenoble, France
8 Polyclinique les Jasmins, Centre d'Aide Médicale à la Procréation, Centre Urbain Nord, Tunis, Tunisia
9 Service de Médecine de la Reproduction, Centre Hospitalier Universitaire Jean Verdier, Assistance Publique - Hôpitaux de Paris, Bondy, France
10 Univ. Grenoble Alpes/CNRS, TIMC-IMAG, CNRS UMR 5525, Grenoble, France
*Corresponding author. Tel: +33 476 637 408; E-mail: christophe.arnoult@univ-grenoble-alpes.fr
†These authors contributed equally to this work as senior authors

*et al*, 2002; Beall *et al*, 2010; Hourvitz *et al*, 2010). We studied a cohort of patients who had all had at least one *in vitro* fertilisation (IVF) cycle yielding only GV, MI or atretic oocytes, and named this phenotype oocyte meiotic deficiency (OMD).

Generation of knockout mouse models has allowed the identification of several genetic variants linked to oocyte meiotic arrest at various stages. For instance, mice deficient in Cdc25b, a gene involved in cyclic AMP control, show GV arrest (Lincoln *et al*, 2002; Vaccari *et al*, 2008). Similarly, deletion of H1foo, a transcription factor for Mei1 (required for normal meiotic chromosome synapsis) and Ubb (a ubiquitin controlling the destruction of key cell cycle regulators), resulted in MI arrest (Libby *et al*, 2002; Furuya *et al*, 2007; Ryu *et al*, 2008). Finally, invalidation of Smc1b, a meiosis-specific component of the cohesin complex, causes MII arrest (Takabayashi *et al*, 2009) while deletion of Mlh3, which maintains homologous chromosome pairing at meiosis, induces mixed arrests (Lipkin *et al*, 2002). While it is tempting to suggest that mutation of any of the above-mentioned genes could cause OMD in women, none has so far been associated with this disease. Recently, heterozygous mis-sense mutations in *TUBB8*, an oocyte-specific tubulin required to form the meiotic spindle, were identified in a cohort of Chinese patients with OMD (Feng *et al*, 2016). Thus, *TUBB8* was established as the first human gene linked to OMD.

Here, we analysed 23 unrelated OMD patients from North Africa and found that six (26%) had the same homozygous truncating mutation in the *PATL2* gene, encoding a putative oocyte-specific RNA-binding protein. The role of this protein has yet to be characterised in mammals. A *TUBB8* variant was only found in a single patient in our cohort, indicating that absence of PATL2 is the main cause of OMD in this region.

## Results

### A homozygous truncating mutation in *PATL2* identified by whole exome and Sanger sequencing in 26% of tested subjects

We analysed a cohort of 23 infertile women presenting with OMD (Table 1). These patients responded normally to ovarian stimulation, and the number of follicles and oocytes harvested was similar to numbers for control patients. However, examination of the oocytes revealed only either GV or MI-arrested or atretic cells (identified by an irregular shape with a dark ooplasm), and a complete absence of MII oocytes.

Given that most of the patients were Tunisian and that 20–30% of marriages are consanguineous in this country, we hypothesised that infertility could be transmitted through recessive inheritance and we therefore focused on homozygous mutations. Exome analysis was performed first on samples from 15 patients. After exclusion of common variants and application of technical and biological filters (Coutton *et al*, 2018), three genes were found to be homozygously mutated in at least two subjects. Only one gene carried a homozygous variant scored as "high" and was predicted to induce loss of function by the "Variant Effect Predictor" tool (Ensembl). Interestingly, the same variant, p.Arg160Ter, c.478C>T in *PATL2* transcript ENST00000434130, was detected in five different patients. Since the orthologue of PATL2 in *Xenopus* is

described as an important factor in *Xenopus* oocyte maturation (Nakamura *et al*, 2010), it was possible that this variant could be the cause of these subjects' infertility. The variant identified is expected to lead to either the production of a truncated protein (Fig 1A) or a complete absence of expression due to possible nonsense-mediated mRNA decay. The truncated protein would contain less than one-third of the complete amino acid sequence, in particular lacking the topoisomerase II-associated protein PAT1 domain. This domain, common to all Pat1 proteins, has been shown to be necessary for its paralogue, PATL1, to function through interaction with its partners (Braun *et al*, 2010) (Fig 1A). Because we did not have access to the relevant biological material (patients' ovaries), it was impossible to assess RNA decay in the presence of this mutation.

The presence of the genetic variant was confirmed by Sanger sequencing for the five mutated patients (Fig 1B). This variant was also identified in a heterozygous state in five out of 148,732 alleles (rs548527219) in the Genome Aggregation Database (gnomAD). This rate corresponds to a very low frequency of 0.003362%, compatible with recessive transmission of a genetic disease. Sanger sequencing of *PATL2* coding sequences was then performed on another eight OMD subjects. An additional patient was identified with the same homozygous mutation, increasing the final number to six out of 23 subjects analysed (26%) carrying the *PATL2* p.Arg160Ter variant.

To complete the analysis of the cohort, WES was performed on the newly recruited patients (*n* = 8) except for the subject harbouring the PATL2 mutation. WES analysis was therefore performed on a total of 22 subjects. From these data, we also sought *TUBB8* heterozygous mutations, which have also been described to induce OMD (Feng *et al*, 2016). One deleterious heterozygous variant (ENST00000309812.4:c.363_366del, ENSP00000311042.4:p.Lys122ArgfsTer13) was identified in patient P16 (Table 1), which could be the reason for this patient's infertility.

Overall, in this cohort, six out of 23 subjects analysed (26%) were observed to carry the *PATL2* p.Arg160Ter mutation, and one patient presented a new *TUBB8* variant (4.5%, 1/22), the pathogenicity of which remains to be confirmed. In our cohort, we compared patient characteristics between subjects with a *PATL2* mutation or presenting no *PATL2* mutation (Fig EV1). Although both groups were of similar ages at the time of analysis, and the numbers of oocytes retrieved were comparable, the two groups were clearly distinct in terms of the type of oocyte arrest. Oocytes from *PATL2* patients were mainly arrested at the GV stage, whereas oocytes from non-*PATL2* patients were generally arrested at the MI stage (Fig EV1).

During evaluation of our data, PATL2 gene mutations were also reported to be associated with OMD in two Asian studies based on cohorts from China and Saudi Arabia (Chen *et al*, 2017; Maddirevula *et al*, 2017). These findings support the causality of our *PATL2* variant and indicate a wide global spread for *PATL2*-dependent OMD.

### Patl2 is not expressed in the hypothalamic–pituitary–gonadal axis in mice

x-pat1a, the *Xenopus* orthologue of *PATL2*, has been reported to be specifically expressed in growing oocytes (Marnef *et al*, 2010;

**Table 1.  Medical history, laboratory investigations and oocyte collection outcomes for patients presenting with OMD.**

| | Origin | Age (years) | Number of oocytes collected | | | | | FSH U/L | LH U/L | TSH U/L | Prolactin µg/L | Menst. | Comments |
|---|---|---|---|---|---|---|---|---|---|---|---|---|---|
| | | | GV | MI | MII | At. | Tot. | | | | | | |
| Patients with *PATL2* mutation | | | | | | | | | | | | | |
| P1 | Tunisia | 35 | 4 | 0 | 0 | 1 | 5 | | | 1.36 | | | |
| | | 34 | | | | | 2 | | | | | | Medical records not available |
| | | 34 | | | | | 8 | | | | | | Medical records not available |
| P2 | Tunisia | 28 | 9 | 0 | 0 | 11 | 20 | | | | | YES | |
| | | 28.9 | 15 | 0 | 0 | 4 | 19 | | | | | | |
| P3 | Tunisia | 24 | 11 | 0 | 0 | 5 | 16 | 10.31 | 3.54 | | 22.28 | | 1 GV maturated to M1 *in vitro* |
| P4 | Tunisia | 34.28 | 8 | 0 | 0 | 2 | 10 | | | 1.07 | 23 | YES | |
| P5 | Arab | 41 | 2 | 0 | 0 | 2 | 4 | 9.39 | 6.1 | 2.3 | 12.9 | YES | |
| | | 42 | 3 | 1 | 0 | 1 | 5 | | | | | | |
| P6 | Mauritania | 36 | 2 | 10 | 0 | 4 | 16 | 3.01 | 3.38 | 2.88 | 25 | YES | Cytoplasmic vacuoles in MI oocyte |
| | | 36.8 | 0 | 0 | 0 | 5 | 5 | | | | | | |
| Patients without *PATL2* mutation | | | | | | | | | | | | | |
| P7 | Algeria | 37 | 2 | 0 | 0 | 2 | 4 | 10.1 | 9.15 | 2.3 | | YES | |
| P8 | Algeria | 32 | 0 | 4 | 0 | 0 | 4 | | | | | | |
| | | 32 | 0 | 2 | 0 | 0 | 2 | | | | | | |
| P9 | Tunisia | 32 | 0 | 0 | 0 | 8 | 8 | | | 3.73 | | | First cousin couple |
| P10 | Tunisia | 37 | 0 | 2 | 0 | 3 | 5 | | | | | | |
| | | 37 | 2 | 3 | 0 | 3 | 8 | | | | | | |
| P11 | Tunisia | 38.9 | 0 | 3 | 2 | 2 | 7 | 8.49 | 3.42 | 1.17 | 17.05 | | |
| P12 | Libya | 28 | 2 | 15 | 0 | 0 | 17 | 1.8 | | | 9.6 | | |
| P13 | Arab | 33 | 0 | 3 | 2 | 7 | 12 | | | | | | |
| | | 37 | 3 | 1 | 0 | 11 | 15 | | | | | | |
| P14 | Tunisia | 26 | 0 | 0 | 0 | 7 | 7 | | | | | YES | |
| P15 | Arab | 38 | 0 | 0 | 0 | 4 | 4 | | | | | | |
| | | 39 | 0 | 0 | 0 | 5 | 5 | | | | | | |
| P16 | Arab | 33 | 0 | 5 | 2 | 2 | 9 | 4.65 | 2.71 | | 14.25 | | Heterozygous mutation in *TUBB8* |
| P17 | Arab | 34 | 0 | 0 | 0 | 0 | 0 | | | | | YES | |
| P18 | Arab-FR | 27 | 0 | 8 | 0 | 4 | 12 | | | | | | |
| P19 | Arab-FR | 24 | 0 | 3 | 0 | 0 | 3 | | | | | | |
| P20 | | 29 | 0 | 0 | 0 | 7 | 7 | | | | | | |
| P21 | Tunisia | 29 | 0 | 0 | 0 | 13 | 13 | | | | | YES | |
| P22 | Tunisia | 39 | 0 | 10 | 0 | 0 | 10 | | | | | | |
| P23 | Tunisia-FR | No data available | | | | | | | | | | | No fertilisation |
| | Mean P1–P23 | 33.42 | 2.17 | 2.50 | 0.21 | 3.90 | 8.45 | | | | | | |
| | Mean P1–P6 | 34.00 | 6.00 | 1.38 | 0 | 3.89 | 10.00 | | | | | | |

**Table 1** (continued)

| | Origin | Age (years) | Number of oocytes collected | | | | | FSH U/L | LH U/L | TSH U/L | Prolactin µg/L | Menst. | Comments |
|---|---|---|---|---|---|---|---|---|---|---|---|---|---|
| | | | GV | MI | MII | At. | Tot. | | | | | | |
| | Mean P7-P23 | 33.1 | 0.45 | 2.95 | 0.3 | 3.9 | 7.6 | | | | | | |
| | Control cohort values (n = 238) | 34.4 | 2.2 | 1.8 | 6 | 2.3 | 9.1 | < 10.2 | < 16.9 | 0.5–5 | 2–20 | | |

Normal values correspond to couples where the male suffers from azoospermia or teratozoospermia ($n = 234$). Arab-FR = French of Arab origin, At. = atretic, Tot. = total, Menst. = menstruation.

Nakamura *et al*, 2010), and analysis of publicly accessible data banks shows that *PATL2* is also expressed at high levels in both human and mouse oocytes (Appendix Fig S1), indicating an important role for PATL2 in female gametogenesis. It should be noted that PATL2 expression is very low in human follicular cells (Appendix Fig S1), suggesting that the maturation defect is of oocyte rather than follicular origin. It also appears that PATL2 is expressed at low levels in a number of other tissues (Appendix Fig S1). We therefore wondered whether an element of the infertility phenotype could be caused by alteration of the hypothalamic–pituitary–gonadal axis. To address this question, we performed comparative Western blots on extracts from GV oocytes, hypothalamus and pituitary glands from Patl2-HA-tagged mice created using CrispR-cas9 technology. Whereas a clear and specific signal is observed for oocyte extracts from PATL2-HA females, no signal was observed in extracts from the hypothalamus or pituitary gland indicating that the direct control of the hypothalamus/pituitary gland on oocyte maturation is not altered in mice (Appendix Fig S2). Since our *PATL2* patients exhibited normal hormone levels (when data were available, Table 1) and reported regular menstrual cycles, these results taken together suggest that the human infertility phenotype is purely due to an oocyte defect.

### Absence of Patl2 modifies the number of the primordial follicles at 26 dpp but not at 12 dpp

To decipher the molecular pathogenesis of the phenotype observed in our *PATL2* patients, we assessed the reproductive phenotype of Patl2-deficient mice ($Patl2^{-/-}$). The gene was invalidated by insertion of a LacZ cassette and deletion of exon 7, inducing a downstream translational frameshift (Appendix Fig S3). The putative transcript produced from this construct would consist of the first 102 amino acids (out of 529) tethered to β-galactosidase. Even if a protein product was generated from the modified Patl2 gene, it would not contain the topoisomerase II-associated protein (PAT1) domain and would therefore not be functional.

Initially, we performed a comparative histological study of control and $Patl2^{-/-}$ ovaries at 12 and 26 days postpartum (dpp). At 12 dpp, ovary sections from control and $Patl2^{-/-}$ females revealed no differences in the mean number of primordial, primary and secondary follicles (Appendix Fig S4). These data indicate that Patl2 plays a marginal role during the development of embryonic ovaries. At 26 dpp, a similar number of primary and secondary follicles were also observed in ovary sections

(Fig EV2A and B). However, there was an unexpected increase in the number of primordial oocytes per section in $Patl2^{-/-}$ ovaries (Fig EV2B). To take the range of secondary follicle sizes into account and to assess follicle growth, we compared histograms plotting the amplitude of follicle diameter for secondary follicles between $Patl2^{-/-}$ and control animals and found no difference (Fig EV2C).

### Patl2 may play a major role in oocyte growth: it is expressed in oocytes from the primary follicle stage and is less abundant from the late GV stage

We used PATL2-HA mice to characterise the function of Patl2 in mouse oocytes. The HA tag was selected for its small size (nine amino acids) making it is less likely to induce tag-dependent relocalisation. Homozygous Patl2-HA females' fertility parameters are well within the normal range for this strain in natural mating (first litters 24 days after crossing with males and litter sizes of 7 and 8, $n = 2$), implying that the HA tag does not impair the function of Patl2. We performed IF and confocal microscopy to quantify the expression of Patl2-HA in the different stages of GV oocytes, in MII oocytes and in ovary sections from tagged mice. The specificity of the fluorescence signal was first validated by immunostaining WT ovary sections alongside Patl2-HA sections using the same anti-HA antibody. No signal was observed in any oocyte in control sections (Appendix Fig S5). In Patl2-HA ovary sections, primordial follicle oocytes produced no detectable fluorescence signal (Fig 2). Signal intensity increased, interpreted as an increase in protein concentration, in oocytes from primary to pre-antral secondary follicles, becoming weaker in oocytes contained in tertiary/antral follicles. It should be noted that fluorescence intensity is not a direct measure of total protein quantity, since oocyte volume increases as follicular stages progress. Because the volume of antral follicle oocytes is larger than that of pre-antral follicle oocytes (approx. 70 versus 50 µm), the total quantity of Patl2 in antral follicle oocytes remains greater, indicating that production of Patl2 is continuous during oocyte growth (Fig 2A–C).

We next compared Patl2-HA immunostained GV oocytes at various stages, and MII oocytes obtained after hormonal stimulation. GV oocytes from pre-antral and antral follicles were obtained by collagenase treatment and ovarian puncture, respectively. In agreement with our results on ovarian sections, the strongest fluorescence signal was observed in pre-antral (secondary) follicle oocytes. GV oocytes from antral follicles can be divided into two categories

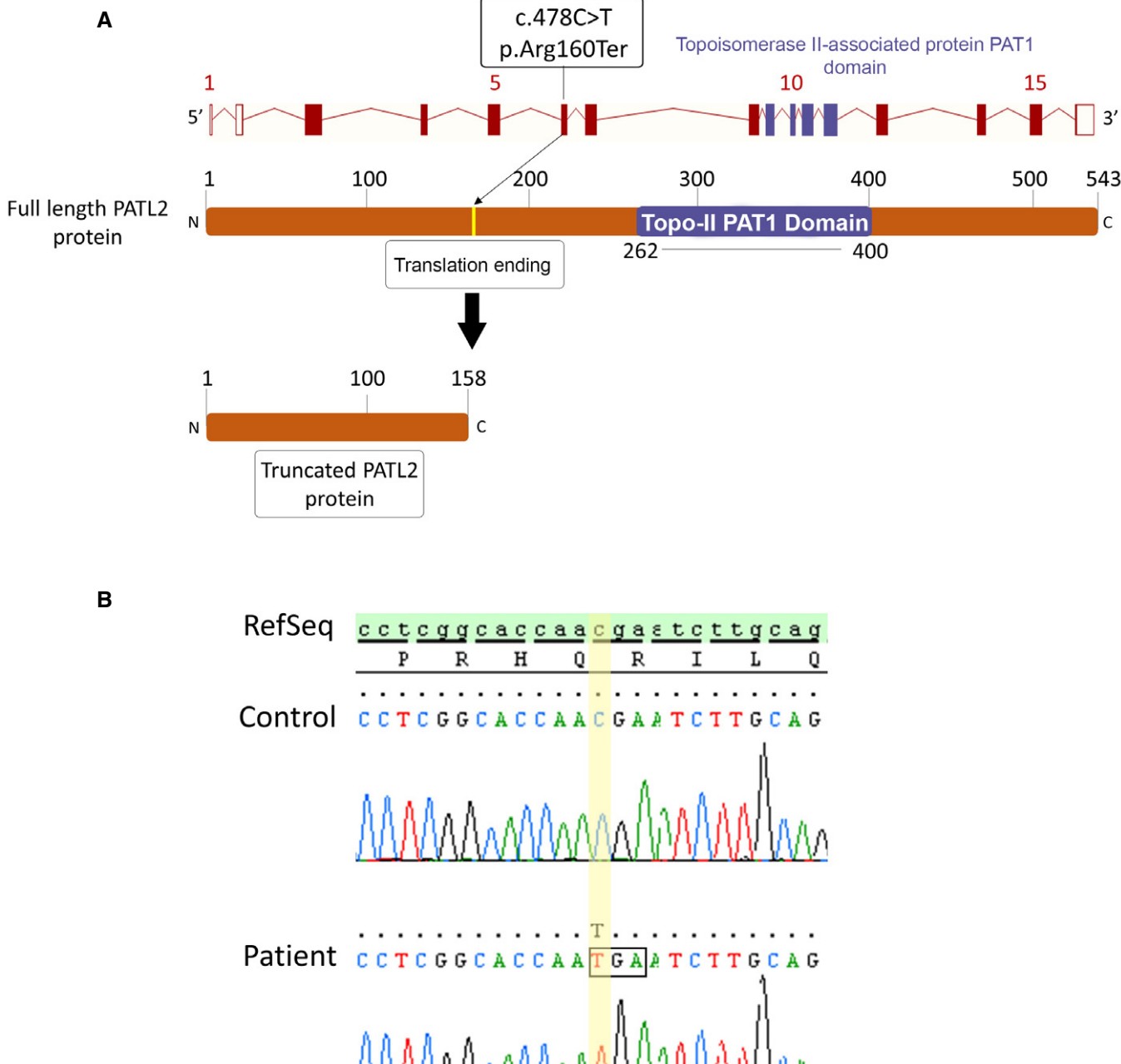

**Figure 1.   Identification of a truncating mutation in *PATL2*.**

A   Location of the *PATL2* mutation in the intron–exon structure and in a representation of the corresponding amino acid sequence. The variant identified, homozygous in the six patients, is located in exon 6 and creates a STOP codon, ending translation and producing a truncated 158-amino acid (aa) protein instead of the full-length 543 aa, and lacking the essential PAT1 (topoisomerase II-associated protein PAT1) domain.

B   Electropherograms of Sanger sequencing for patients harbouring *PATL2* mutations compared to reference sequence.

based on the nuclear distribution of their chromatin: the non-surrounded nucleolus (NSN) to surrounded nucleolus (SN) conformational change occurs in the final stages of GV oocyte development and correlates with transcriptional arrest (De La Fuente, 2006). Stronger Patl2-HA fluorescence was observed in NSN than in SN GV oocytes (Fig 2D). PATL2-HA was also detected in MII oocytes at a level comparable to that in SN GV oocytes (Fig 2D).

During oocyte growth, a large quantity of stable mRNA necessary for growth and maturation accumulates within the oocyte. Up to 30% of this mRNA is translationally repressed until meiotic

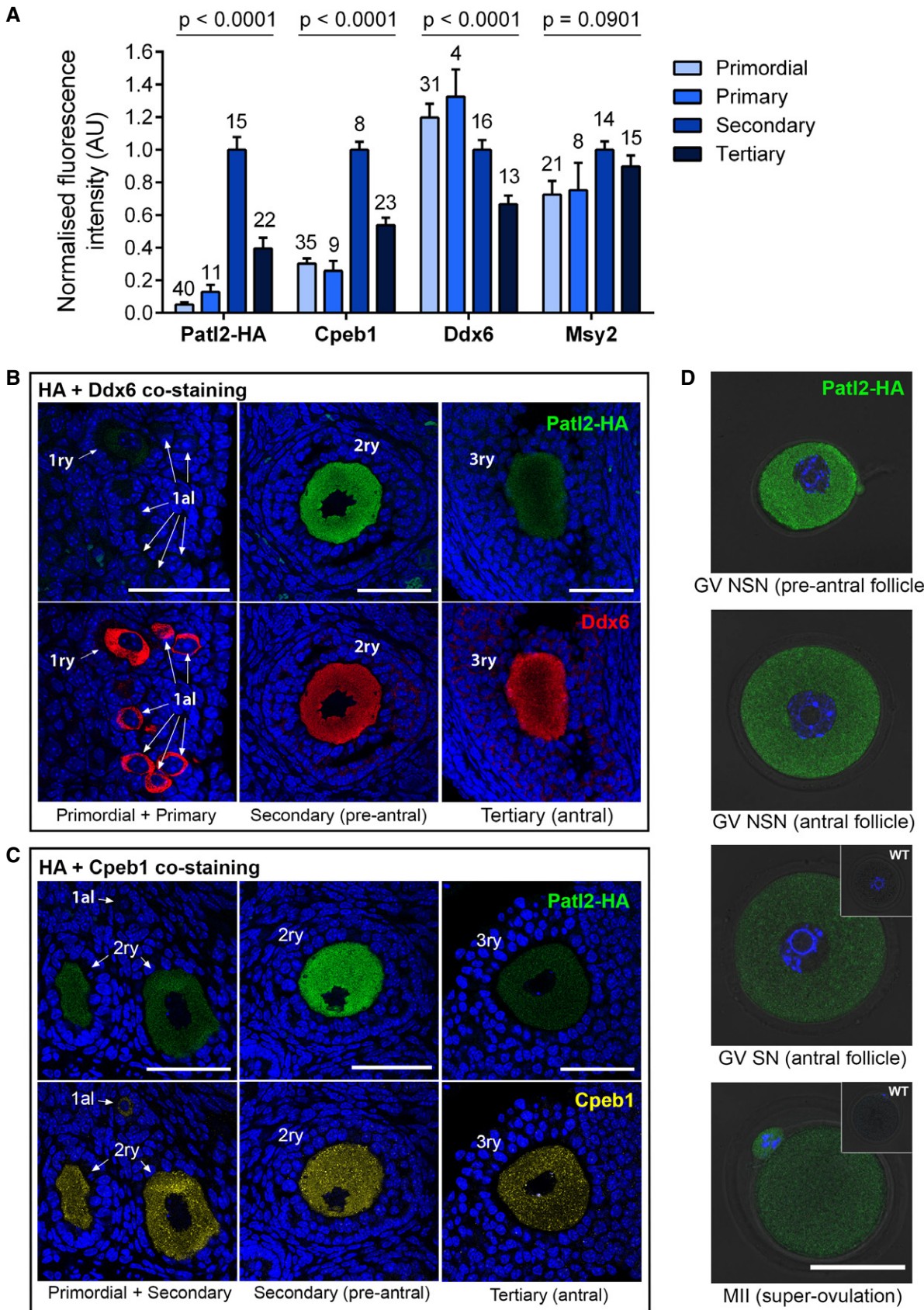

Figure 2.

◄

**Figure 2. Patl2, Cpeb1, Ddx6 and Msy2 expression profiles during oocyte growth and meiotic maturation.**

A   Ovary sections (3 μm thick) from Patl2-HA homozygous mice were co-stained with antibodies against the HA tag and either Cpeb1, Ddx6 or Msy2. Normalised mean fluorescence intensity at different follicle stages (as indicated) was measured using confocal microscopy. The mean fluorescence intensity of secondary follicle oocytes was used to normalise intensities for each protein to take into account variations in overall staining intensity between slides and obtain comparable values. Numbers above bars correspond to the size of the sample. Data are presented as mean ± SEM. Statistical differences were determined based on ANOVA test, *P*-value as indicated.

B   Variation in fluorescence intensity for Patl2 and Ddx6 during oocyte growth. Confocal images obtained from the same section of an ovary from a Patl2-HA female, co-stained with antibodies against HA tag and Ddx6. Patl2 staining is not detectable in primordial oocytes, barely detectable in primary oocytes, and has a maximum intensity in secondary oocytes. Note that Ddx6 staining is strong in primordial oocytes. Primordial (1al), primary (1ry), secondary (2ry) follicles are indicated. Sections were counterstained with Hoechst to reveal the nucleus. Scale bar = 50 μm.

C   Variation in fluorescence intensity for Patl2 and Ddx6 during oocyte growth. Confocal images obtained from the same section of an ovary from a Patl2-HA female, co-stained with antibodies against HA tag and Cpeb1. Note that Cpeb1 is detectable in primordial oocytes and that its staining is more punctiform than Patl2 staining, with numerous foci observable at all stages. Primordial (1al), primary (1ry), secondary (2ry) follicles are indicated. Sections were counterstained with Hoechst to reveal the nucleus. Scale bar = 50 μm.

D   Comparative Patl2 staining of GV oocytes from pre-antral, antral with NSN (non-surrounded nucleolus) chromatin, antral with SN (surrounded nucleolus) chromatin and MII oocytes from Patl2-HA-tagged mice. GV oocytes were isolated by ovarian puncture, and MII oocytes were collected in the oviduct from stimulated Patl2-HA females. After fixation, oocytes were stained with anti-HA antibody and observed by confocal microscopy. In GV SN and MII oocytes, insets correspond to WT oocytes at the same developmental stage, showing no fluorescent staining. Scale bar = 50 μm.

maturation or after fertilisation. RNA processing involves very large complexes of ribonucleoproteins (RNPs), which are involved in the storage, processing, regulation and/or degradation of mRNA, and whose function often depends on the protein's phosphorylation status and the protein composition of the RNP. Clusters of RNP complexes are known as P-bodies. In *Xenopus* oocytes, x-Pat1a binds to the cytoplasmic polyadenylation element binding complex (CPEB), a central RNP involved in RNA translation/storage (Marnef *et al*, 2010; Nakamura *et al*, 2010). We therefore wondered whether other proteins known to be RNP components expressed in mouse oocytes, such as Cpeb1, Msy2 (thought to be x-Pat1a partners) and Ddx6 (a crucial P-body component) presented similar patterns of expression/abundance to Patl2 during oocyte growth (Flemr *et al*, 2010; Medvedev *et al*, 2011). To answer this question, we quantified these proteins in oocytes at the different follicular stages by IF and confocal microscopy (Fig 2A). Cpeb1 showed a similar, but not identical, pattern of intensities to that observed for Patl2. The main difference was that Cpeb1 is expressed in primordial follicle oocytes (in which Patl2 is undetectable). As for Patl2, the intensity was highest in secondary follicle oocytes and weakened in tertiary follicle oocytes (Fig 2B). Unlike Patl2, Msy2 and Ddx6 were abundantly expressed in primordial follicle oocytes, and their fluorescence intensity varied little throughout oocyte growth (Fig 2C and also Fig EV2 for Msy2).

We next assessed possible colocalisation between Patl2 and Cpeb1, Msy2 and Ddx6 (Fig 3). These experiments were performed on ovarian sections of secondary/pre-antral follicles, where the strongest Patl2 signal was recorded. Cpeb1 and ddx6 present a clear punctiform signal (Fig 3: A4, B4), unlike Msy2, for which the signal is more homogenous (Fig 3: C4). The Patl2-HA signal can be described as a homogenous scattering of small dots (Fig 3: A3, B3, C3). The dots are clearly smaller in size than those corresponding to Cpeb1 and Ddx6 staining, and no obvious colocalisation between Patl2 and any of the three proteins was observed (Fig 3: A2, B2, C2).

### Subfertility in *Patl2*⁻/⁻ female mice is due to compromised oocyte maturation and poor developmental competence of oocytes and embryos

We next assessed fertility in *Patl2*⁻/⁻ animals by crossing them with WT animals and counting the number of live pups per litter, the total number of live pups born and the number of litters per month

over a 6-month period. *Patl2*⁻/⁻ females exhibited severe subfertility: the number of pups per litter dropped from 7.3 ± 0.8 ($n = 14$ litters for three females) for WT to 2.3 ± 0.4 ($n = 7$ litters for three females) for *Patl2*⁻/⁻ mice (Fig 4A and B), and both the total number of pups and of litters per month per female were reduced (Fig 4C and D). Conversely, *Patl2*⁻/⁻ males showed normal fertility: no difference in litter size was observed compared to WT (7.6 ± 0.7, $n = 17$ litters for five WT males and 7.6 ± 0.5, $n = 28$ litters for five *Patl2*⁻/⁻ males) (Fig 4E).

Next, ovarian stimulation was performed and oocytes were collected for morphological and IVF studies. Stimulation with pregnant mare serum gonadotropin (PMSG) was used to obtain GV oocytes with well-defined nucleoli from both WT and *Patl2*⁻/⁻ females. However, GV oocytes from *Patl2*⁻/⁻ mice were smaller in diameter than those from WT mice, suggesting that the absence of Patl2 affected oocyte growth (Fig EV3A and B). Full stimulation (PMSG + hCG) produced a comparable number of oocytes in *Patl2*⁻/⁻ and WT mice (Fig EV3C). This result concurs with observations that patients harbouring *PATL2* mutations produce a comparable number of oocytes to control patients (Table 1). *Patl2*⁻/⁻ mice produced MII stage oocytes (Fig EV3D), identified by the presence of the first polar body (PB1), indicating that the phenotype is not as severe in mice as in humans, where no MII oocytes were produced (Table 1). As for GV oocytes, MII *Patl2*⁻/⁻ oocytes were smaller in diameter than control MII oocytes (Fig EV3E), indicating that oocyte meiotic maturation as well as oocyte growth was impaired in the absence of Patl2. This finding was corroborated by the increased percentage of oocytes released at stages before MII: 26% for WT and 45% for *Patl2*⁻/⁻ (Fig 5A). These oocytes were probably blocked in metaphase I (MI), as indicated by the absence of PB1 (Fig 5B). Notably, many apparent MI-arrested oocytes presented misaligned chromosomes and abundant cytoplasmic asters (Fig 5B). A significant increase in morphological defects such as abnormal spindle morphology, misalignment of chromosomes on the spindle and numerous cytoplasmic asters was also observed in *Patl2*⁻/⁻ MII oocytes (Fig 5C and D). Thus, both women carrying a *PATL2* mutation and *Patl2*⁻/⁻ mice exhibit oocyte maturation defects.

Next, the developmental competence of the oocytes collected was challenged in IVF experiments. We chose not to denude oocytes for these experiments since removal of cumulus cells has a negative impact on fertilisation. For WT females, the percentage of eggs

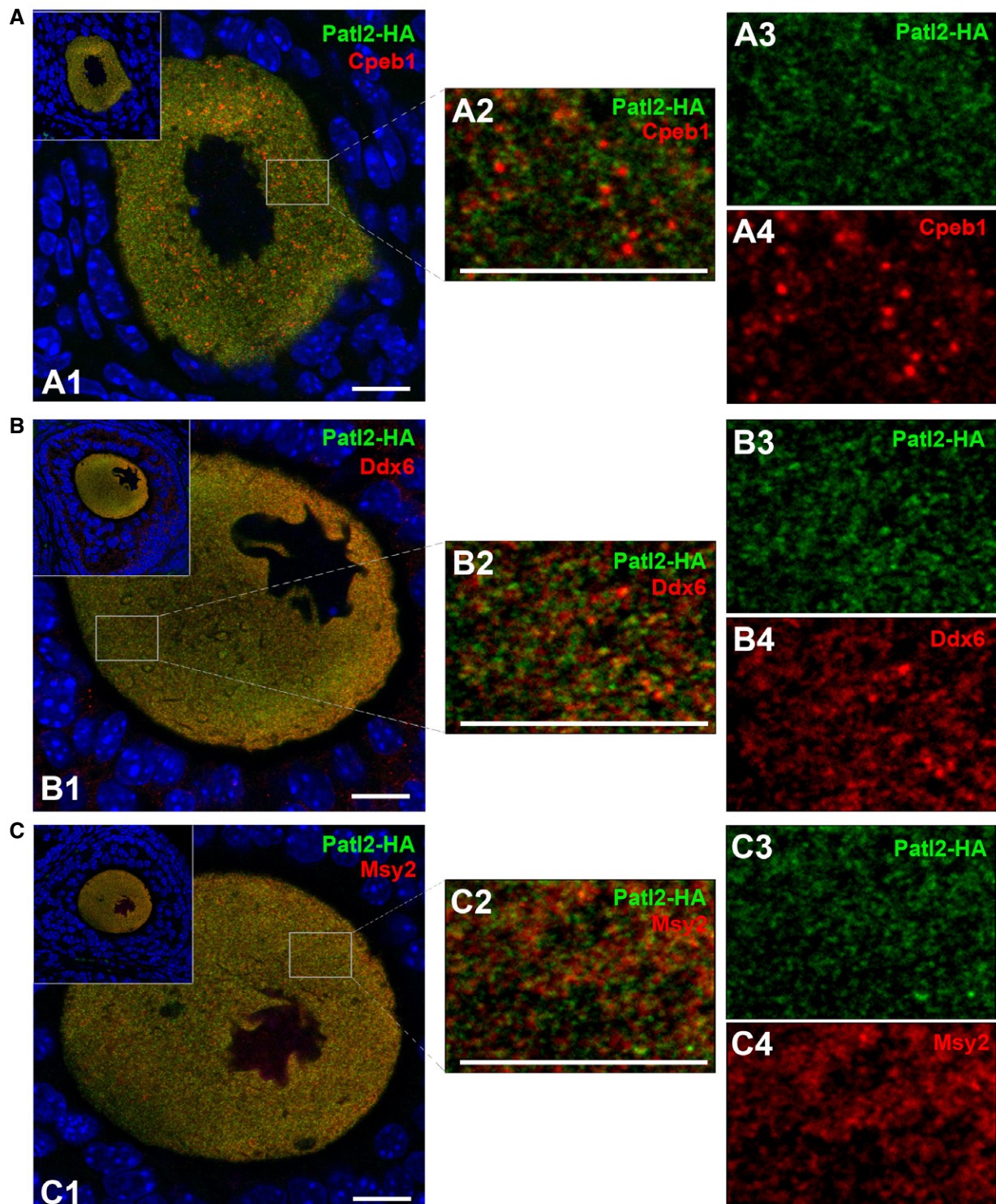

**Figure 3. Analysis of possible colocalisation of Patl2 with Cpeb1, Ddx6 and Msy2.**

A  Confocal image of secondary follicle oocytes from Patl2-HA homozygous females co-stained with antibodies against HA tag (green) and Cpeb1 (red) and counterstained with Hoechst to reveal the nucleus (A1). Insets show the follicular environment of the oocyte studied. White rectangles indicate the zones of enlargement, shown on the right (A2). This image corresponds to a merge of Patl2 (A3) and Cpeb1 (A4) signals. Scale bars = 10 μm.

B  Similar experiments performed with secondary follicle oocytes from Patl2-HA homozygous females co-stained with antibodies against HA tag (green) and Ddx6 (red). Scale bars = 10 μm.

C  Similar experiments performed with secondary follicle oocytes from Patl2-HA homozygous females co-stained with antibodies against HA tag (green) and Msy2 (red). Scale bars = 10 μm.

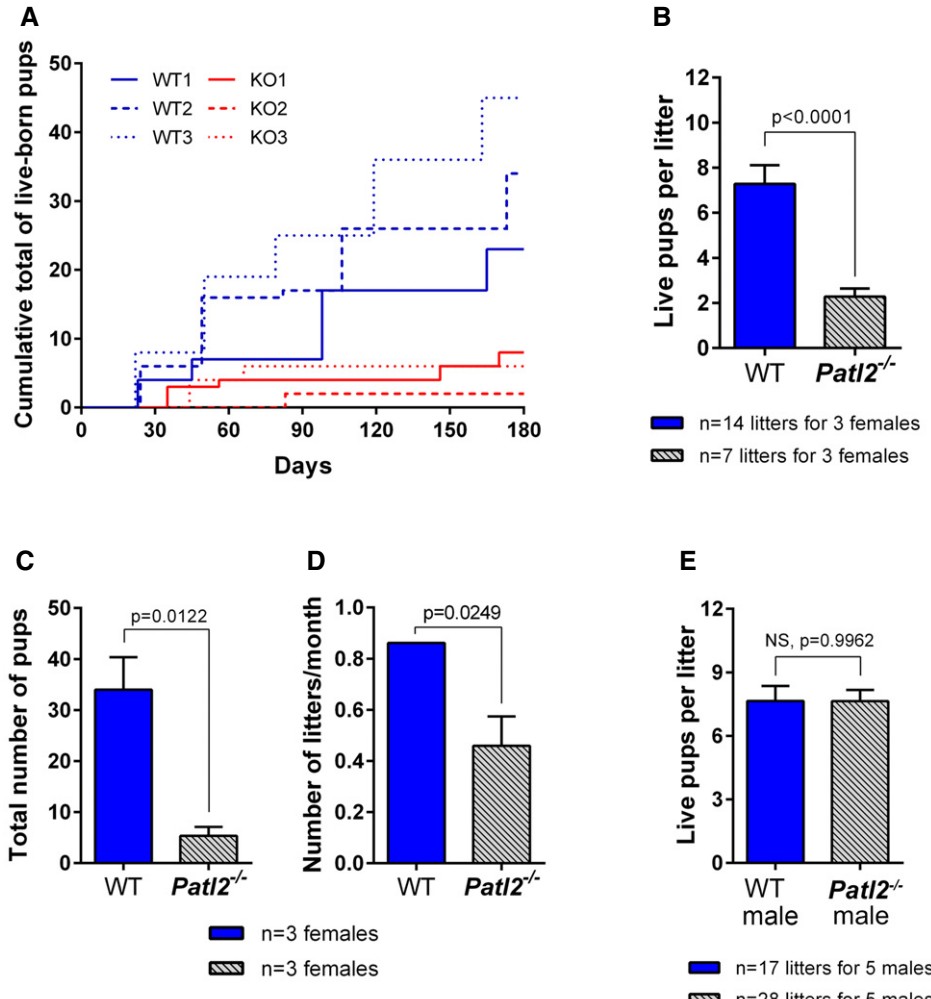

**Figure 4. *Patl2* knockout (*Patl2*$^{-/-}$) females exhibit a severe subfertility phenotype when mated with WT males, whereas *Patl2*$^{-/-}$ males are fertile.**

A  Comparative accumulation of live pups over a period of 6 months from three WT and three *Patl2*$^{-/-}$ females crossed with WT males shows severe hypofertility of *Patl2*$^{-/-}$ females.

B  Histograms showing the number of pups per litter (mean ± SEM) obtained by crossing three WT (*n* = 14 total litters) and three *Patl2*$^{-/-}$ females (*n* = 7 total litters) with WT males. Statistical test: two-tailed unpaired *t*-test with Welch's correction.

C  Total number (mean ± SEM) of pups produced by a WT or *Patl2*$^{-/-}$ female over a 6-month period (*n* = 3 per genotype). Statistical test used: two-tailed unpaired *t*-test.

D  The number of litters per month (mean ± SEM) is significantly decreased for *Patl2*$^{-/-}$ versus WT females. Statistical test: two-tailed unpaired *t*-test.

E  *Patl2*$^{-/-}$ males produce comparable litter sizes to WT males when mated with WT females (mean ± SEM). Statistical test: two-tailed unpaired *t*-test.

reaching the two-cell stage was 67.7% ± 8.1 (*n* = 5 experiments, 10 females) (Fig 6A). Given that only 74% of ovulated oocytes can be assumed to be at the MII stage (Fig 5A), this proportion translates to 90% of WT MII oocytes reaching the two-cell stage. In contrast, for *Patl2*$^{-/-}$ females the percentage of eggs reaching the two-cell stage dropped to 36.4% ± 6.4, which translates to 65% success if we consider that only 55% of ovulated oocytes from *Patl2*$^{-/-}$ females are MII oocytes (Fig 5A). The IVF outcomes were therefore significantly altered in *Patl2*$^{-/-}$ females, indicating compromised developmental competence for *Patl2*$^{-/-}$ oocytes. This decrease in numbers of two-cell *Patl2*$^{-/-}$ embryos correlates with the abnormal development of *Patl2*$^{-/-}$ zygotes, which exhibited numerous defects, including delayed pronucleus formation, absence of sperm DNA decondensation and/or

polyspermy. In contrast, almost all fertilised WT zygotes contained two pronuclei (2PN) (Fig 6B–D). Finally, the reduced developmental competence of *Patl2*$^{-/-}$ eggs and their altered fertilisation also severely affected pre-implantation development since only 27.2% ± 5.1 (*n* = 4 experiment, eight females) of two-cell embryos generated with *Patl2*$^{-/-}$ eggs reached the blastocyst stage in contrast to 87.1% ± 5.6 with WT eggs (Fig 6A).

**Absence of Patl2 significantly alters the transcriptome of GV and MII oocytes**

Since x-Pat1a, the *Xenopus* orthologue of *PATL2,* is a RNA-binding protein (Marnef *et al*, 2010; Nakamura *et al*, 2010), we next

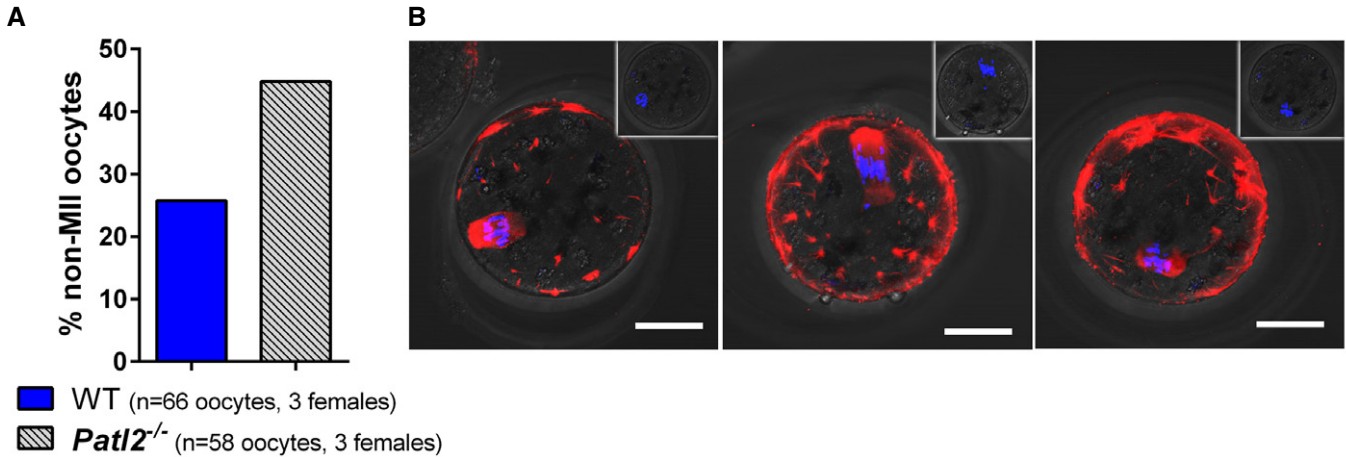

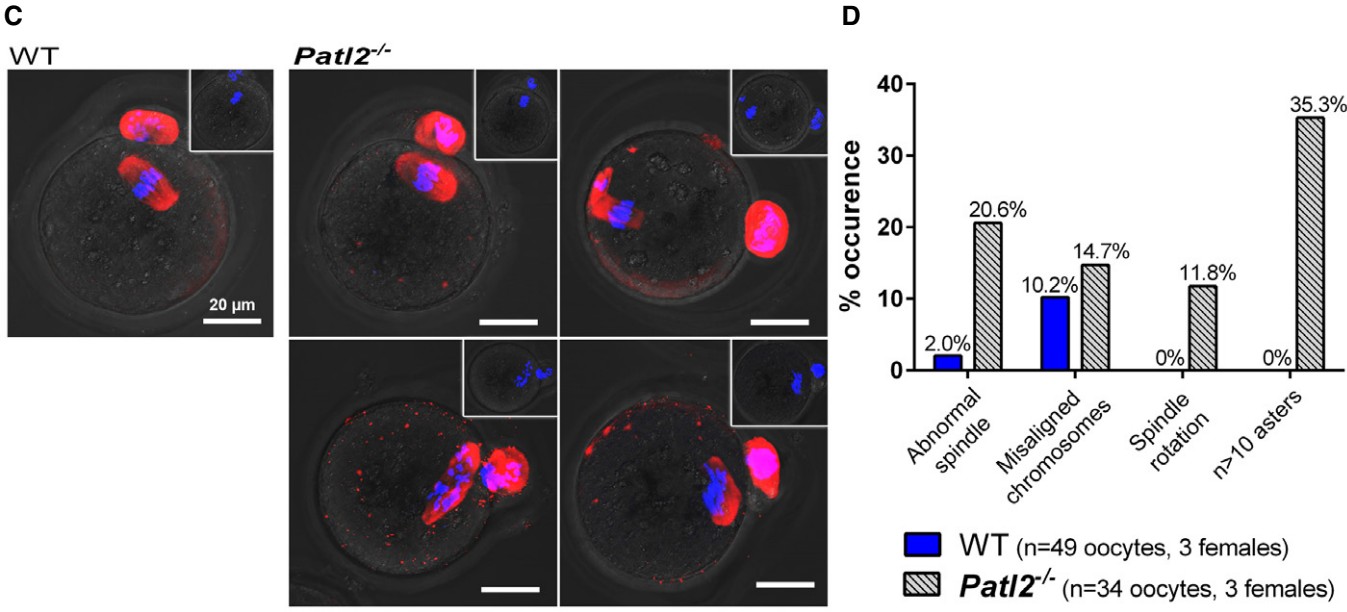

**Figure 5. Infertility of *Patl2*-deficient female mice is due to oocyte maturation defects.**

A   Oocytes collected after ovarian stimulation were labelled with a tubulin antibody (red) and counterstained with DAPI to reveal DNA (blue). An increase in non-MII oocytes (MI arrest, absence of PB) after full ovarian stimulation was observed in *Patl2*$^{-/-}$ mice (*n* = 3 females per genotype).

B   IF images of tubulin-stained *Patl2*$^{-/-}$ oocytes arrested at MI stage showing various defects such as irregular spindle shape and abnormal chromosome distribution. Scale bar = 20 μm. Inset in each panel shows overlay of phase contrast image and Hoechst staining of the corresponding oocyte. No polar bodies were observed for MI oocytes.

C   IF images of tubulin-stained WT and *Patl2*$^{-/-}$ MII oocytes, as evidenced by PB1. In control MII oocytes, stack projections of confocal images show that the spindle was symmetric and the chromosomes distributed in the middle of the spindle. In contrast, in *Patl2*$^{-/-}$ MII oocytes various defects were observed such as irregular spindle shape, spindle rotation and numerous cytoplasmic asters. Slightly greater numbers of oocytes with abnormal chromosome distribution were also observed. Scale bar = 20 μm. Inset in each panel shows overlay of phase contrast image and Hoechst staining of the corresponding oocyte. One polar bodies was observed for MII oocytes.

D   Histograms quantifying the % defects observed in *Patl2*$^{-/-}$ MII oocytes.

assessed how the absence of Patl2 affected the oocyte transcriptome during oocyte maturation in mice. To do so, global gene expression analysis was performed on oocytes collected at GV and MII stages from WT and *Patl2*$^{-/-}$ females.

Expression levels for nearly 66,000 transcripts were measured across the different oocyte groups using Affymetrix microarrays.

First, we verified that oocyte RNA purification was not contaminated by RNA from follicular cells, by comparing expression levels of genes specific to follicular cells and oocytes, respectively (Appendix Fig S6A). The absence of exon 7 transcription in *Patl2*$^{-/-}$ oocyte extracts was also verified in the microarray data (Appendix Fig S6B). The results of this analysis revealed no

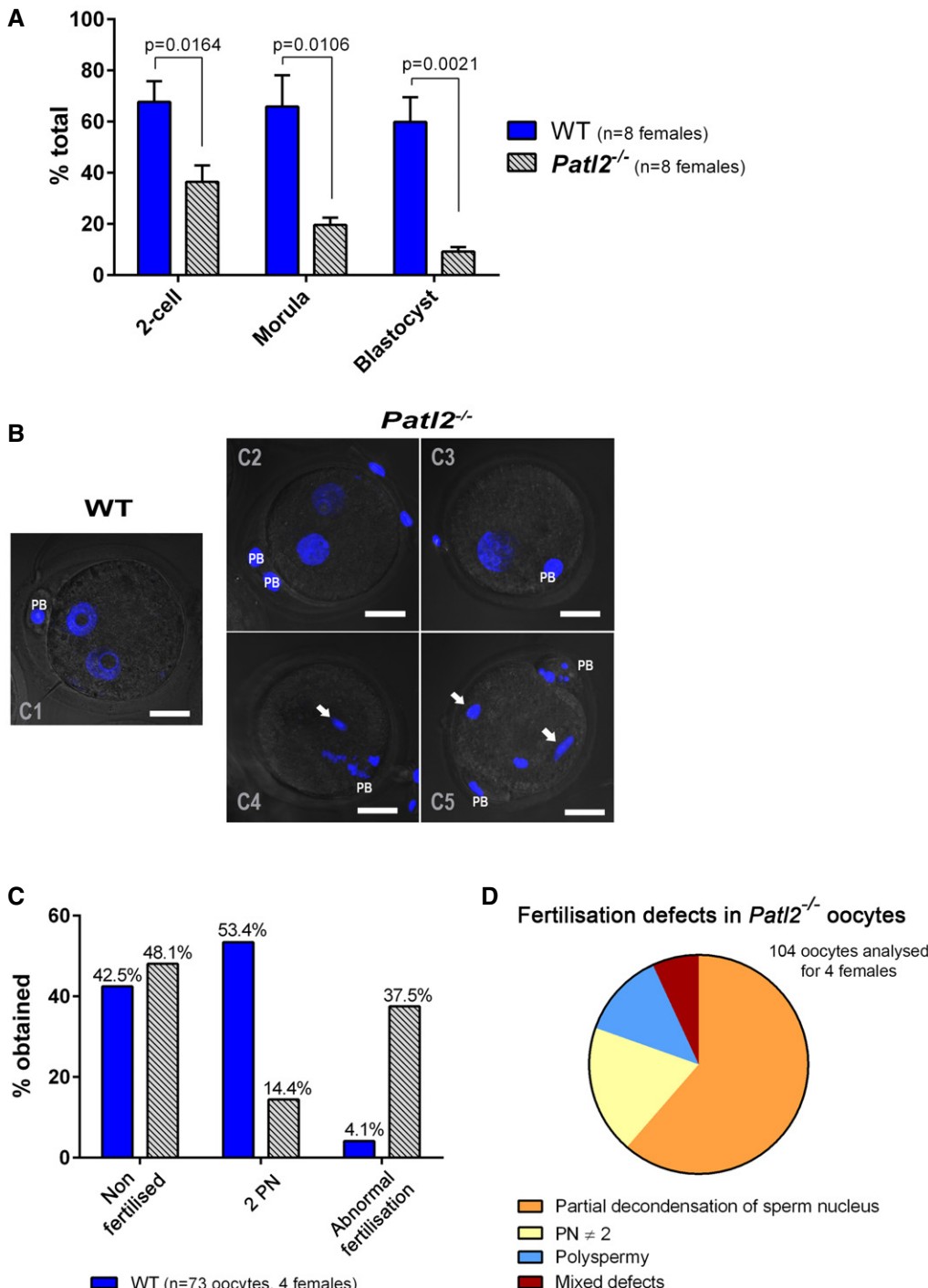

**Figure 6.  MII oocytes from *Patl2*⁻/⁻ female mice exhibit abnormal fertilisation preventing normal embryo development.**

A       IVF outcomes (mean ± SEM) measured at the two-cell, morula and blastocyst stages show that the developmental competence of *Patl2*⁻/⁻ oocytes is compromised. Oocytes were collected from stimulated WT and *Patl2*⁻/⁻ females and sperm from WT males. For each IVF replicate (different WT males), IVF outcomes at different stages were compared for WT and *Patl2*⁻/⁻ oocytes. Statistical differences were assessed using unpaired two-tailed *t*-tests.

B       *Z*-stack projections of confocal images of 2PN zygotes obtained from WT and *Patl2*⁻/⁻ oocytes 6–8 h after sperm–egg mixing. Fertilised WT oocytes exhibit normal 2PN stage (C1) whereas the number of fertilised *Patl2*⁻/⁻ oocytes exhibiting normal 2PN stage is strongly reduced (C2) and most of them show defects such as absence of a male pronucleus (C3), partial decondensation of male PN (C4, C5, white arrows) or polyspermy (C5 white arrows). Scale bar = 20 μm, PB = polar body, PN = pronucleus.

C, D    The percentage of 2PN obtained 6–8 h after fertilisation drops from 53.4% for WT to 14.4% for *Patl2*⁻/⁻ oocytes, which exhibit various fertilisation abnormalities including partial decondensation of sperm DNA, polyspermy, abnormal number of PN or mixed defects (D).

difference in overall purified RNA concentration between WT and $Patl2^{-/-}$ oocytes, for both GV and MII stages, indicating that unlike Msy2, Patl2 is not a global regulator of mRNA stability and translation (Yu *et al*, 2001) (Appendix Fig S6C).

However, significant changes were observed for specific transcripts, both at GV and MII stages (Fig 7A). At the GV stage, lack of Patl2 induced a > twofold decrease for 95 transcripts ($P < 0.05$) and > twofold increase for 39 transcripts (Dataset EV1). At the MII stage, a > twofold decrease in 124 transcripts and > twofold increase in 122 transcripts were observed (Dataset EV2). Approximately one-third of the genes down-regulated at the GV stage (32) were also down-regulated at the MII stage, and half of the genes up-regulated at the GV stage (19) were also up-regulated at the MII stage (Dataset EV3 and Fig 7B). The impact of Patl2 deletion on gene expression at the GV and MII stages was then visualised using hierarchical clustering of genes with an absolute fold-change ((aFC)$> 2$, $P < 0.05$ (Fig 7C).

## Phosphorylation, oxidation and other pathways are down-regulated in the absence of Patl2

Literature mining to determine the functions of the genes down-regulated in $Patl2^{-/-}$ oocytes identified several groups of genes reported to be involved in oocyte maturation. The proteins encoded are involved in several signalling pathways implicated in oocyte differentiation, oxidative stress, transcription and translation, exocrine modulation, meiosis and spindle formation (Table 2). We therefore suspect that the decreased expression of these genes interferes with normal oocyte differentiation. Regarding signalling pathways that are activated during oocyte maturation, significant down-regulation of transcripts for proteins involved in mTORC1, Wnt, NF-kB, MAP kinase and phosphatase signalling pathways was observed. It is worth noting that we also detected a more than twofold decrease in Pgrmc1, a receptor required to slow down oocyte meiotic progression (Guo *et al*, 2016; Table 2). This decrease in Pgrmc1 in GV oocytes was confirmed by RT–qPCR experiments (Appendix Fig S7).

Oocyte maturation is controlled by bidirectional crosstalk between the follicular cells and the oocyte: the secretion of oocyte-produced factors is necessary for follicle cell differentiation; follicle cells in turn secrete factors activating different signalling pathways within the oocyte (Li & Albertini, 2013). Interestingly, two factors, Cxcl14 and Adm2, known to play a crucial role in cumulus cell maturation (Bobe *et al*, 2006; Chang *et al*, 2011), were down-regulated in $Patl2^{-/-}$ oocytes (Table 2). We also observed a strong deregulation of several other interesting transcription factors, such as Sohlh2, which was down-regulated 2.2-fold and 5.3-fold at the GV and MII stages, respectively, and Eef1e1 (Table 2). We also noticed that two glutathione-S-transferases were repressed in $Patl2^{-/-}$ GV oocytes (Table 2). This down-regulation may increase

oxidative stress within the oocyte. Spindle defects may also be aggravated by repression of Pak4 and Ccdc69 (Table 2), two proteins known to affect spindle assembly (Pal *et al*, 2010; Bompard *et al*, 2013). RT–qPCR experiments confirmed the down-regulation of Ccdc69 and Eef1e1 in GV oocytes (Appendix Fig S7).

Some transcripts, such as *Fgf9* and *Cdc25a* which control meiosis II (Assou *et al*, 2006), are known to be strongly up-regulated in oocytes after GVBD. Interestingly, both these transcripts were significantly down-regulated in $Patl2^{-/-}$ MII oocytes (Table 2). This decreased expression probably hampers final oocyte maturation. Among the list of genes up-regulated at both GV and MII (Dataset EV3), two particular genes, *Prr11* and *Ska2* (spindle and kinetochore associated complex subunit 2), are known to be required for spindle stability (Zhang *et al*, 2012). Ska2 up-regulation was confirmed by RT–qPCR for GV oocytes (Appendix Fig S7).

Finally, the lists of all down- and up-regulated proteins in $Patl2^{-/-}$ oocytes at the GV and MII stages were uploaded directly into Ingenuity Pathway Analysis (IPA) software, to explore their molecular and biological functions. At the GV stage, the most significantly down-regulated pathways were the ephrin receptor pathway and ephrin A signalling, which control embryo/trophoblast development (Fig EV4A and Table 2). Interestingly, at the MII stage, the cell cycle G1/S checkpoint was also significantly down-regulated (Fig EV4B), in accordance with the cell cycle defects visible in Fig 5.

## The transcripts differentially expressed between MII and GV in WT and $Patl2^{-/-}$ samples affect a similar set of biological pathways and functions

To assess a possible role for Patl2 in the GV-MII transition, we identified the up- or down-regulated differential expression transcripts (DET) ($P < 0.05$, absolute fold-change > 2) between the GV and MII transcript lists for WT and $Patl2^{-/-}$ samples (Datasets EV4 and EV5). The lists of corresponding proteins were directly uploaded into IPA software to explore their molecular and biological functions. The functional pathways or networks with the highest confidence scores were then determined by right-tailed Fisher's exact tests. This analysis showed that a large portion of DET in WT samples were involved in energy production (oxidative phosphorylation and mitochondrial (dys)function, Appendix Fig S8) protein synthesis (EIF2 signalling, regulation of eIF4 and p7056K signalling) and DNA replication, recombination and repair (Fig EV5A). It is worth noting that the up- and down-regulated transcripts included in the DET lists were documented to functionally interact with each other, forming tightly connected networks. This observation strengthens the relevance of the data (Appendix Fig S9A). A similar analyse of DET from $Patl2^{-/-}$ samples indicated that the eight pathways with highest confidence scores were identical to those recorded for WT samples (Fig EV5B), suggesting that Patl2 only plays a minor role (or no role) in the

---

**Figure 7.  Transcriptional analysis of GV and MII oocytes from WT and Patl2$^{-/-}$ mice.**

A   Comparison of the transcriptional profiles in $Patl2^{-/-}$ oocytes versus WT oocytes at the GV or MII stages. GV oocytes were collected 44 h after PMSG injection and MII 13 h after hCG injections. For both MII and GV, two replicates for WT and three replicates for $Patl2^{-/-}$ oocytes were analysed.

B   Venn diagrams representing down- or up-regulated genes in $Patl2^{-/-}$ oocytes (absolute fold-change ((aFC))> 2, $P < 0.05$) with respect to WT oocytes at the GV and MII stages.

C   Hierarchical clustering of gene expression data for the down- and up-regulated genes (aFC> 2, $P < 0.05$) of $Patl2^{-/-}$ and WT oocytes at GV (left) and MII (right) stages, demonstrating the clustering of replicates to their respective groups.

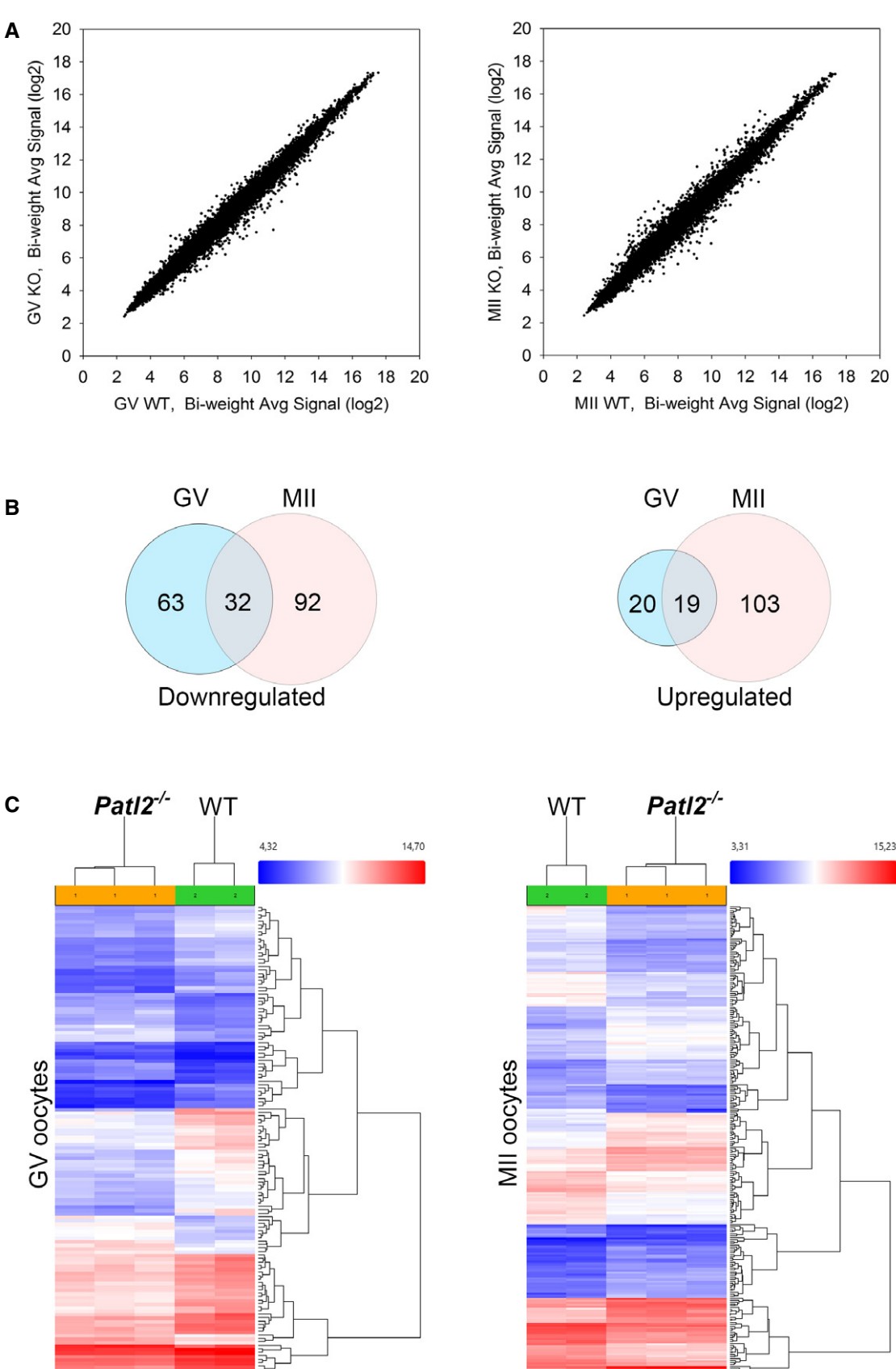

**Figure 7.**

**Table 2. Gene ontology analysis of microarray data.**

| Gene symbol | Gene name | WT GV | KO GV | Fold-change | WT MII | KO MII | Fold-change | (Possible) role in oocyte maturation and early embryo development | References |
|---|---|---|---|---|---|---|---|---|---|
| Elements of signalling pathways activated during oocyte maturation | | | | | | | | | |
| Gatsl3 | GATS protein-like 3 | 9.64 | 8.12 | −2.88 | 9.26 | 8.99 | −1.21 | CASTOR (gatsl3) is an arginine sensor for the mTORC1 pathway. Arginine breaks the CASTOR/GATOR2 complex. GATOR2 activates mTORC1. Activated mTORC1 will phosphorylate translation inhibitor 4E-BP1, releasing it from eukaryotic translation initiation factor 4E (eIF4E) which is then free to join the translation initiation complex. When the mTORC pathway is inhibited, bovine oocytes are blocked at the M1 stage | Chantranupong et al (2016), Mayer et al (2014) |
| Pygo1 | Pygopus 1 | 8.68 | 6.9 | −3.43 | 8.25 | 6.73 | −2.87 | Pygopus is a Wnt transcriptional component. Knockdown of Axin-1, a negative regulator of Wnt signalling leads to defective spindles, misaligned chromosomes, PB1 extrusion failure and impaired PN formation. Embryo development is also impacted since maternal Wnt/STOP signalling promotes cell division during early Xenopus embryogenesis. WNT signalling pathway is important for proper oocyte maturation | He et al (2016), Huang et al (2015), Spate et al (2014) |
| Ppp1r14b | Protein phosphatase 1, regulatory (inhibitor) subunit 14B | 12.62 | 11.11 | −2.84 | 12.65 | 11.34 | −2.49 | PP1 is an important protein involved in the cell cycle and controls dephosphorylation of numerous proteins such as proteins phosphorylated by Cdc2 and downstream mitotic kinases. Likely impacts meiotic control | |
| Ppp1r15a | Protein phosphatase 1, regulatory (inhibitor) subunit 15A | 8.26 | 7.29 | −1.96 | 9.12 | 8.17 | −1.92 | | |
| Dusp1 | Dual specificity phosphatase 1 | 11.94 | 10.49 | −2.72 | 12.78 | 11.78 | −2.01 | Dusp1 is able to dephosphorylate MAP kinase, known to be very important in oocyte maturation and meiosis | Liang et al (2007) |
| Nfkbia | Nuclear factor of kappa light polypeptide gene enhancer in B cells inhibitor, alpha | 9.78 | 8.33 | −2.73 | 9.43 | 7.7 | −3.31 | NFKBIA is a gene involved in maintaining meiotic transcriptional arrest. Inhibits the NF-κB transcription factor. Change during GV-MII transition. Highly expressed at embryonic genome activation | Paciolla et al (2011) |
| Pgrmc1 | Progesterone receptor membrane component 1 | 9.16 | 7.97 | −2.27 | 9.53 | 8.01 | −2.86 | P4−PGRMC1 interaction attenuated oocyte meiotic progression and primordial follicle formation by decreasing intra-oocyte cAMP levels. When PGRMC1 is low, oocytes are not blocked by P4 and mature too fast | Guo et al (2016) |
| Secreted factors | | | | | | | | | |
| Cxcl14 | Chemokine (C-X-C motif) ligand 14 | 8.08 | 6.99 | −2.13 | 8.59 | 7.5 | −2.13 | Important factor for oocyte maturation in fish | Bobe et al (2006) |

**Table 2** (continued)

| Gene symbol | Gene name | WT GV | KO GV | Fold-change | WT MII | KO MII | Fold-change | (Possible) role in oocyte maturation and early embryo development | References |
|---|---|---|---|---|---|---|---|---|---|
| Adm2 | Adrenomedullin 2 | 11.75 | 10.64 | −2.16 | 12.4 | 11.91 | −1.4 | ADM2 is a novel oocyte-derived ligand important for the regulation of cell interactions in COCs that functions, in part, by suppressing cumulus cell apoptosis | Chang *et al* (2011) |
| Transcription translation factors | | | | | | | | | |
| Med11 | Mediator complex subunit 11; | 9.84 | 8.28 | −2.94 | 9.78 | 8.39 | −2.62 | Mediator of RNA polymerase II transcription, subunit 11 homologue (*S. cerevisiae*) | |
| Med29 | Mediator complex subunit 29 | 11.01 | 9.33 | −3.2 | 7.97 | 7.32 | −1.57 | Mediator complex subunit 29 (MED29) is part of a large multiprotein coactivator complex that mediates regulatory signals from gene-specific activators to general transcription machinery in RNA polymerase II mediated transcription | |
| Sohlh2 | Spermatogenesis and oogenesis specific basic helix-loop-helix 2 | 7.18 | 6.06 | −2.18 | 8 | 5.58 | −5.33 | SOHLH2 is a transcription factor that coordinates oocyte differentiation without affecting meiosis I and drives oocyte growth and survival but not meiosis I | Choi *et al* (2008), Shin *et al* (2017) |
| Slx2 | Sycp3 like X-linked | 8.04 | 6.81 | −2.36 | 5.97 | 5.77 | −1.15 | SLX2 might be involved in DNA recombination, synaptonemal complex formation as well as sex body maintenance during meiosis | Shi *et al* (2013) |
| Prmt5 | Protein arginine N-methyltransferase 5 | 9.05 | 7.88 | −2.25 | 8.72 | 8.16 | −1.47 | PRMT5 negatively affects cyclin E1 promoter activity. Cyclin E1 is activated during meiosis in *Xenopus*. Prmt5 is critical in biological function in a wide range of cellular processes including development and methylates histones H2A and H4 in oocytes | Stopa *et al* (2015), Wilczek *et al* (2011) |
| Eef1e1 | Eukaryotic translation elongation factor 1 epsilon 1 | 13.39 | 12.26 | −2.19 | 13.24 | 11.78 | −2.76 | DNA damage response. Regulator of translation | Uniprot |
| E2f4 | E2F transcription factor 4 | 13.61 | 12.5 | −2.17 | 14.02 | 13.18 | −1.8 | Regulator of translation | |
| Oxidative stress | | | | | | | | | |
| Gstp2 | Glutathione S-transferase, pi 2 | 12.04 | 10.41 | −3.11 | 10.28 | 9.51 | −1.71 | Important in oxidative stress. A high level of gluthatione transferase is associated with higher oocyte developmental quality. Glutathione S-transferase is a marker of oocyte maturity in pigs | Paczkowski and Krisher (2010), Rausell *et al* (2007) |
| Gstp1 | Glutathione S-transferase, pi 1 | 11.43 | 9.91 | −2.86 | 9.74 | 9.14 | −1.51 | | |
| Spindle assembly | | | | | | | | | |
| Pak4 | p21 protein (Cdc42/Rac)-activated kinase 4 | 11.32 | 10.32 | −2.01 | 12.02 | 11.41 | −1.52 | Ran is a substrate for p21-activated kinase 4 (PAK4). RanGTP is an important actor of spindle formation and asymmetric division during meiosis | Bompard *et al* (2013) |

**Table 2**  (continued)

| Gene symbol | Gene name | WT GV | KO GV | Fold-change | WT MII | KO MII | Fold-change | (Possible) role in oocyte maturation and early embryo development | References |
|---|---|---|---|---|---|---|---|---|---|
| Ccdc69 | Coiled-coil domain containing 69 | 14.63 | 13.54 | −2.13 | 14.8 | 13.87 | −1.91 | CCDC69 regulates central spindle assembly | Pal et al (2010) |
| Embryo/trophoblast factors | | | | | | | | | |
| Phlda2 | Pleckstrin homology-like domain, family A, member 2 | 10.3 | 7.56 | −6.7 | 7.7 | 7.05 | −1.57 | PHLDA2 is an imprinted gene, and only the maternal copy is expressed. This gene is associated with placental dysfunction. KO mice exhibit foetal growth deficiency | Frank et al (2002), Jensen et al (2014) |
| Efna1 | Ephrin A1 | 9.73 | 7.34 | −5.24 | 9.05 | 8.02 | −2.04 | Ephrins A1-4 were expressed in blastocysts. The ephrin A system is involved in regulating contact between blastocysts and endometrium during embryo implantation | Fu et al (2012), Fujii et al (2006) |
| Efna4 | Ephrin A4 | 8.7 | 7.08 | −3.07 | 8.55 | 7.21 | −2.54 | | |
| Efna2 | Ephrin A2 | 8.9 | 7.49 | −2.66 | 8.81 | 7.86 | −1.93 | | |
| Cstb | Cystatin B | 11.81 | 9.44 | −5.17 | 10.85 | 9.5 | −2.55 | The cathepsin–cystatin system plays an important role in trophoblast cell invasion and normal embryonic growth | Nakanishi et al (2005) |
| Prl8a2 | Prolactin family 8, subfamily a, member 2 | 9.51 | 7.17 | −5.05 | 9.12 | 5.93 | −9.16 | In Prl8a2 null tissues, genes expressed in the trophoblast are down-regulated | Alam et al (2015) |
| Crabp2 | Cellular retinoic acid binding protein II | 12.73 | 10.5 | −4.69 | 12.34 | 10.73 | −3.06 | Altered expression level of endometrial CRABP2 is involved in abnormal endometrium-trophoblast interaction, which leads to implantation failure | Lee et al (2011) |
| Tlr8 | Toll-like receptor 8 | 7.34 | 5.49 | −3.61 | 7.15 | 5 | −4.46 | TLR8 is expressed in the trophoblast. Its inhibition suggests that it is necessary for successful establishment of early gestation in ewes | Kaya et al (2017), Ruiz-Gonzalez et al (2015) |
| Specific MII regulation | | | | | | | | | |
| Fgf9 | Fibroblast growth factor 9 | 6.64 | 6.1 | −1.45 | 7.21 | 5.88 | −2.51 | FGF9 counteracts retinoic acid, which promotes entry into meiosis. Its expression increases at MI stage and reaches highest level at the MII stage | Feng et al (2014) |
| Cdc25a | Cell division cycle 25A | 11.59 | 11.55 | −1.03 | 12.31 | 11.18 | −2.19 | Its expression increases significantly at the MII stage. Cdc25a is crucial in the MI-MII transition and its down-regulation results in fewer oocytes resuming meiosis and reaching MII | Solc et al (2008) |

GV-MII transition. Similarly, in $Patl2^{-/-}$ samples, DET formed tightly connected networks (Appendix Fig S9B).

# Discussion

### Genetic basis of OMD

The genetic basis of OMD was unexplored until 2016 when heterozygous mutations of *TUBB8* were reported to account for approximately 30% (7/23) of OMD Han Chinese familial cases (Feng et al, 2016). In our 22-subject cohort, only one *TUBB8* deletion variant was identified by WES analysis, suggesting that *TUBB8* mutations may not be a common cause of OMD in North African women. No down-regulation of the expression of tubulin genes was observed in GV or MII oocytes from $Patl2^{-/-}$ mice, indicating that *PATL2*-dependent OMD does not involve tubulin deregulation. These results suggest geographical heterogeneity of the genetic causes of OMD. Indeed, 26% of our patients were found to present the same homozygous truncation of *PATL2*, demonstrating that *PATL2* mutation is a major cause of OMD

in North Africa. Interestingly, the same mutation was detected in all our patients, suggesting a founder effect. This hypothesis was reinforced by the analysis of the variants from WES data in the regions surrounding *PATL2*, which revealed a common homozygous haplotype (Dataset EV6). During the assessment of our data, two other studies were published associating *PATL2* with human OMD. The first was carried out on a cohort of 180 Han Chinese patients: it identified 5 *PATL2* nonsense variants and four mis-sense variants in five unrelated subjects. If the mis-sense variants are proven to be deleterious, this corresponds to a frequency of ~3% in the subjects tested (5/180) (Chen *et al*, 2017). In the second study, two homozygous *PATL2* variants were identified in two OMD subjects each from two different Saudi Arabian families. One of these variants is the same p.Arg160Ter variant presented here, whereas the other is a new mis-sense variant (Maddirevula *et al*, 2017). With cases across two continents, these studies, in combination with the present study, provide strong evidence that *PATL2* mutation is a major cause of human OMD whose phenotype is characterised by arrested development at the germinal vesicle stage.

**Comparison of expression profiles and distribution for Patl2 and other RNA-binding proteins suggest a niche role for Patl2**

By IF, we showed that (i) Patl2 is undetectable in primordial follicle oocytes; (ii) significant Patl2 translation begins at the primary follicle stage; (iii) Patl2 concentration peaks in secondary follicle oocytes; (iv) Patl2 is expressed only in the cytoplasm; (v) staining is relatively homogenous; and (vi) taking oocyte volume into account, the total amount of Patl2 increases as oocytes grow. These results clearly indicate that Patl2 plays a key role during oocyte growth and that it does not contribute to the maintenance of the primordial follicle pool in mouse ovaries. This conclusion is corroborated by our data showing comparable numbers of primordial follicles in control and Patl2$^{-/-}$ ovaries at 12 dpp. Nevertheless, the number of primordial follicles was found to be significantly higher at 26 dpp in Patl2$^{-/-}$ ovaries. This puzzling result deserves further investigation; it may indicate that expression of Patl2 is required for the primordial/primary follicle transition. The importance of Patl2 in oocyte growth is reinforced by our data showing that GV and MII oocytes are smaller in diameter in the absence of Patl2. Interestingly, although signal intensity for Patl2 declined during the GV NSN to SN transition, the protein remained clearly detectable in MII oocytes. This observation contrasts with data from *Xenopus* oocytes, where x-Pat1a was shown to completely disappear upon progesterone-induced meiotic maturation (Marnef *et al*, 2010; Nakamura *et al*, 2010). This discrepancy suggests that the function of Patl2 may have evolved in mammals to include a role during fertilisation and/or early development. A similar evolution has been reported for the germ-cell-specific translational repressor Msy2, which is detectable until the late two-cell embryo stage, undergoing changes in its phosphorylation state which presumably influence its interaction with mRNA (Yu *et al*, 2001).

Oocyte growth is characterised by extensive mRNA synthesis and accumulation within the cytoplasm. Although most of the mRNA synthesised during oocyte growth is immediately translated to support the growing oocyte, up to 30% of mRNAs are stored for subsequent translation and are required at meiotic resumption or for early zygote development (Pique *et al*, 2008). Several molecular mechanisms

controlling mRNA stability have been described and involve a number of RNA-binding proteins (Clarke, 2012), although their relative contributions remain poorly understood. The best-described translational repression mechanism in oocytes involves a specific mRNA sequence, the cytoplasmic polyadenylation element (CPE), which is present in the 3′ UTR of the affected mRNAs. It associates with its binding factor, CPEB, as well as the cytoplasmic polyadenylation machinery (Racki & Richter, 2006; Sousa Martins *et al*, 2016). The role of CPEB in mediating translational repression is well documented, but its precise mechanism of action—in particular the roles of its molecular partners—remains to be fully elucidated. x-Pat1a, for example, was described to be an RNA-binding protein and a partner of CPEB in *Xenopus* oocytes, but its precise role in translational repression remains unknown (Nakamura *et al*, 2010). Apart from CPEB, the RNA-binding protein Msy2 also plays a key role in RNA stability. The *Xenopus* orthologue of Msy2 (Ybx2) has been reported to participate in the same complex as x-Pat1a and CPEB in *Xenopus* oocytes (Medvedev *et al*, 2011). Absence of Msy2 in mice leads to infertility and modulates the expression of around 7,000 transcripts (> twofold absolute change), that is approximately 30% of all transcripts. This broad effect could be explained if it acts as a sequence-independent RNA-binding protein or global regulator of mRNA stability. Finally, P-bodies, central to RNA processing in somatic cells, have been described in mammalian oocytes. The RNA helicase and translational repressor Ddx6 is a key component of all known types of P-body. Several studies have reported the existence of P-bodies containing Ddx6 in immature mouse oocytes, which disappear during oocyte growth, dispersing Ddx6 as well as Msy2 and Cpeb within the cytoplasm (Swetloff *et al*, 2009; Flemr *et al*, 2010). We therefore endeavoured to find out whether Patl2 may be part of a similar, temporary, structure.

To assess the possibility that Patl2 works in synergy with these proteins which are important for mRNA stability, we compared their expression profiles and abundance and performed colocalisation experiments. Patl2 localisation was determined using Patl2-HA mice. Msy2 and Ddx6 proteins were found to be expressed at significant levels in primordial follicle oocytes, whereas Patl2 was undetectable at this stage. The Msy2 concentration appeared constant throughout oocyte growth, and its abundance in antral follicle oocytes is in line with its role in the oocyte-embryo transition (Yu *et al*, 2001). Ddx6 is strongly expressed in primordial and primary follicle oocytes, with signal intensity gradually decreasing in secondary and tertiary follicle oocytes. As Ddx6 is a P-body component, this result is compatible with the disappearance of P-bodies in fully grown oocytes (Flemr *et al*, 2010). Unlike Patl2, Cpeb1 was detectable in primordial follicle oocytes, but its signal intensity pattern was otherwise similar to that of Patl2, peaking in secondary follicle oocytes and weakening in tertiary follicle oocytes. However, Cpeb1 appears to be localised in small RNPs, as evidenced by its clear punctiform staining (Fig 3). In colocalisation experiments performed on secondary follicle stage oocytes, Patl2 failed to clearly colocalise with any of the three mRNA regulatory proteins. Taken together, these results suggest that Patl2 may be involved in a new pathway controlling the stability of specific mRNA. This new pathway merits further study.

**Impact of *Patl2* invalidation on the oocyte transcriptome**

In this study, we assessed the impact of the absence of Patl2 on the transcriptomic profiles of GV and MII oocytes and on the

transcriptomic variation observed at the GV-MII transition. The GV-MII transition had previously been reported to be associated with specific degradation of a large number of transcripts associated with protein synthesis, DNA replication and energy production (Su *et al*, 2007). Our results confirm these previous findings, as the pathways that underwent the most significant changes during the GV-MII transition in WT oocytes were associated with oxidative phosphorylation and mitochondrial (dys)function (energy production), EIF2 and regulation of eIF4 and p7056K signalling (protein synthesis) and DNA repair (Su *et al*, 2007). Su *et al* (2007) also showed that 62 out of 88 transcripts in the oxidative phosphorylation pathways were down-regulated at the GV-MII transition. Here, we report down-regulation of 45 out of 88 transcripts, of which 32 are included in the set reported by Su *et al* (2007). Interestingly, our IPA analysis revealed other pathways that are modulated during the GV-MII transition, such as sirtuin signalling and mTOR signalling. The similarity of our results to previous reports, together with RT–qPCR analysis of selected genes, reinforces the conclusions drawn from the transcriptomic analyses. Importantly, we observed effects on the same major pathways in the GV-MII transition in *Patl2*$^{-/-}$ oocytes and in WT oocytes, suggesting that Patl2 is not involved in this transitional phase.

Our data show that the absence of Patl2 induces transcriptomic deregulation, affecting 135 genes at the GV stage and 248 genes at the MII stage with an absolute fold-change > 2, ($P$ < 0.05). This relatively small list includes genes of highly relevant function (Table 2). Indeed, several of the corresponding proteins, such as Prgmc1 and Slohlh2, are reported to be involved in oocyte maturation. Sohlh2 is of particular interest as it has been described to be required for oocyte growth, and Sohlh2$^{-/-}$ females were found to be infertile (Choi *et al*, 2008). Interestingly, this factor does not affect meiosis I (Shin *et al*, 2017), which is line with the observation that *Patl2*$^{-/-}$ females are able to generate MII oocytes. The factors Cxcl14 and Adm2, known to be necessary for cumulus cell maturation, were also found to be down-regulated. However, no decrease in follicle size or any other visible defects were observed in ovarian sections at 26 dpp, suggesting that any follicular defects are subtle. Importantly, several key proteins protecting cells against oxidative stress were down-regulated (GST forms). Oocytes are very sensitive to oxidative stress, which causes spindle abnormalities (Choi *et al*, 2007) and affects developmental competence (Rausell *et al*, 2007); in addition, glutathione S-transferase has been reported to be a marker of oocyte maturity in pigs (Paczkowski & Krisher, 2010).

Some of the up-regulated transcripts could also interfere with normal oocyte maturation. The most extensively up-regulated gene at the MII stage was *Prr11*. In WT MII oocytes, its expression was low, suggesting that its role in normal oogenesis is minimal or null. *Prr11* deregulation has however been shown to dramatically modify the cell cycle, although how it does so remains unclear (Li *et al*, 2017). Its strong up-regulation may therefore interfere with meiosis or early development. Another remarkable gene up-regulated at both GV and MII is *Ska2* (spindle and kinetochore associated complex subunit 2), which is known to control spindle stability during meiosis (Zhang *et al*, 2012). This up-regulation could explain the numerous defects observed in MI and MII meiotic spindles. Up-regulation of these genes could be a cellular response attempting to address the transcript deregulation induced by Patl2-deficiency. Taken together, these results underline the importance of PATL2 in

the regulation of a specific subset of mRNAs required for the generation of mature oocytes.

Moreover, a considerable number of genes known to be involved in pre-implantation embryonic development were also found to be down-regulated in both GV and MII oocytes from *Patl2*$^{-/-}$ mice. This down-regulation may contribute to the poor pre-implantation developmental competence observed for *Patl2*$^{-/-}$ embryos (Fig 6). These results suggest that Patl2 plays an essential role in maintaining the integrity of a small pool of mRNAs synthesised during oocyte growth and necessary after fertilisation and during early embryo development. This role is compatible with the presence of Patl2 in MII oocytes and its function as a translation repressor, as described for its *Xenopus* homologue (Marnef *et al*, 2010).

In conclusion, we report that PATL2 is a vital player in oocyte growth and maturation, where it regulates the expression of mRNAs encoding proteins crucial for oocyte meiotic progression and early embryonic development, and that its invalidation causes OMD in humans. Unravelling the molecular basis of OMD will help patients by improving diagnosis and our understanding of their disease. This work will also be of tremendous interest to the fast-growing field of clinical *in vitro* oocyte maturation, necessary for the development of a number of applications, such as fertility preservation for cancer patients (in particular for young prepubertal girls; Kim *et al*, 2016) before commencing reprotoxic chemotherapy, or *in vitro* maturation of primary/secondary follicles for patients with premature ovarian failure (Kim, 2012; Yin *et al*, 2016).

# Materials and Methods

### Ethics

After sperm analysis and IVF treatment, performed independently of the research presented in this paper, couples presenting an interesting phenotype (male and/or female infertility) were selected and referred by their physician. The physician explained the study and its objectives before subjects signed an informed consent form in line with the local IRB protocols and the principles of the Declaration of Helsinki. DNA samples from couples of interest were collected and the identity of patients coded in such a manner that subjects could not be identified. All medical records were saved. No specific treatment was given to any female patient for research purposes, and all oocytes collected were used for IVF/ICSI treatment only.

### Patients and biological samples

A total of 23 patients were recruited. All subjects were of North African descent, mainly Tunisia and Algeria, with one patient from Mauritania. All had undergone one or two cycles of ovarian stimulation to allow egg collection for use in IVF. Human DNA samples were obtained from patients consulting for diagnosis and assisted reproductive techniques at the PolyClinique des Jasmins (Tunis, Tunisia) ($n$ = 21) or at the reproductive unit located in Bondy, France ($n$ = 2).

Patients underwent standard controlled ovarian stimulation by the administration of gonadotropins using either the long agonist or the antagonist protocol. Women were given between 150 and 225 IU recombinant follicle-stimulating hormone (Gonal-F; Merck-Serono). Subsequently, ultrasound was performed and follicular response

recorded from day 5 of gonadotropin stimulation. When at least two follicles were ≥ 18 mm in diameter, 6,500 IU or 10,000 IU human chorionic gonadotropin was administered. Oocytes were retrieved 34–36 h later by vaginal ultrasound-guided follicular puncture.

Control cohort: Since 2007, and in parallel to this study on OMD, we also performed a large study on male infertility, involving several hundred couples. Most of these couples had IVF/ICSI treatments, and female partners of infertile men with normal fertility levels were considered as fertile controls. Their characteristics were recorded and used anonymously. These cohorts have already been described in our previous works concerning male infertility (Dieterich *et al*, 2007; Harbuz *et al*, 2011; Ben Khelifa *et al*, 2014; Coutton *et al*, 2018; Kherraf *et al*, 2018).

### Exome sequencing and bioinformatics analysis

Genomic DNA was isolated from saliva using Oragene saliva DNA collection kit (DNAgenotek Inc., Ottawa, Canada). Exome capture was performed using NimblGen SeqCap EZ Kit version 2 (Roche). Sequencing was conducted on Illumina HiSeq2000 instruments with paired-end 76-nt reads. Sequence reads were aligned against the reference genome (hg19) using MAGIC (SEQC/MAQC-III Consortium, 2014). Duplicate reads and reads mapping to multiple locations in the genome were excluded from further analysis. Positions with a sequence coverage of < 10 on either the forward or reverse strand were excluded. Single nucleotide variations (SNV) and small insertions/deletions (indels) were identified and quality-filtered using in-house scripts. The most promising candidate variants were identified using an in-house bioinformatics pipeline which is described in (Coutton *et al*, 2018). Variants with a minor allele frequency > 5% in the NHLBI ESP6500 or in 1,000 Genomes Project phase 1 data sets, or > 1% in ExAC, were discarded. These variants were also compared to an in-house database of 56 control exomes. All variants present in a homozygous state in this database were excluded. Variant Effect Predictor (Ensembl) was used to predict the impact of the selected variants. Only variants affecting splice donor/ acceptor sites or causing frameshift, inframe insertions/deletions, stop gain, stop loss or mis-sense variants were retained, except for those scored as "tolerated" by SIFT (sift.jcvi.org) and as "benign" by Polyphen-2 (genetics.bwh.harvard.edu/pph2). All steps from sequence mapping to variant selection were performed using the ExSQLibur pipeline (https://github.com/tkaraouzene/ExSQLibur).

### SANGER sequencing

Sanger sequencing was carried out using the primers listed in Appendix Table S1A. PCR amplification (35 cycles) was performed with a melting temperature of 60°C. Sequencing reactions were performed using BigDye Terminator v3.1 (Applied Biosystems). Sequence analyses were carried out on ABI 3130XL (Applied Biosystems). Sequences were analysed using seqscape software (Applied Biosystems). The primers used for Sanger verification of *PATL2* mutations in patients are listed in Appendix Table S1A.

### Mice

All animal procedures were performed according to the French and Swiss guidelines on the use of animals in scientific investigations after approval of the study protocol by the local Ethics Committee (ComEth Grenoble No. 318, ministry agreement number #7128 UHTA-U1209-CA) and the Direction Générale de la Santé (DGS) for the State of Geneva. *Patl2*$^{FL/FL}$ mice (C57BL/6NTac-Patl2 < tm1a) were generated by the EUCOMM Consortium (http://www.mousephenotype.org/about-ikmc/eucomm). They were obtained from the Mouse Clinical Institute—MCI, Strasbourg, France. Deletion of exon 7 was obtained by crossing them with adult heterozygous EIIaCre transgenic mice (Lakso *et al*, 1996), obtained from Institut Cochin—Inserm 1016—CNRS 8104—Paris. The ellaCre carries a Cre transgene under the control of the adenovirus EIIa promoter. This construction targets expression of Cre recombinase to early mouse embryos. Cre expression is thought to occur prior to implantation in the uterine wall. Cre-mediated recombination thus affects a wide range of tissues, including the germ cells that transmit the genetic alteration to progeny.

Patl2-HA knock-in mice were generated by CRISPR/Cas9 technology (Kherraf *et al*, 2018). Twenty-seven nucleotides encoding the HA tag were inserted in the c-terminal of the *Patl2* gene, immediately before the TAG stop codon (TAC CCA TAC GAT GTT CCA GAT TAC GCT TAG) in C57BL/6 mice. One plasmid containing one sgRNA and Cas9 was injected (5 ng/μL) with the single-stranded DNA (50 ng/μl) for homology-directed repair. The single-stranded DNA was 187 nt long and contained two 80 nt long arms homologous to the nucleotides located before and after the *Patl2* stop codon, surrounding the 27 nucleotides encoding the HA tag, and ending with a TAG stop codon. The sgRNA sequence overlapped the native Patl2 stop codon which, eight nucleotides later, was followed by a TGG representing a suitable protospacer adjacent motif (PAM) sequence. The synthetic ssDNA was purchased from IDT (Leuven, Belgium). After injection, zygotes were left for 4–6 h before introducing them into pseudopregnant host females where they were carried to term. Edited founders were identified by Sanger sequencing from digit biopsies. Mice carrying the desired modification were crossed with C57BL6/J to verify germline transmission and eliminate any possible mosaicism. Heterozygous animals with the same modification were then mated to produce homozygous offspring.

Mice were housed with unlimited access to food and water in a facility with 12-h light per day. Animals were sacrificed by cervical dislocation at the ages indicated.

### Genotyping

DNA for genotyping was isolated from tail biopsies. Tail biopsies (2 mm long) were digested in 200 μl lysis buffer (Direct PCR Lysis Reagent (Tail); Viagen Biotech inc, CA, USA) and 0.2 mg proteinase K for 12–15 h at 55°C, followed by 1 h at 85°C to inactivate proteinase K. The DNA was directly used to perform PCRs. Multiplex PCR was done for 35 cycles, with an annealing temperature of 62°C and an elongation time of 45 s at 72°C. PCR products were separated by 2% agarose gel electrophoresis. Genotypes were determined depending on the migration pattern. Primers for Patl2-KO and Patl2-HA-tagged mice are listed in Appendix Table S1B,C.

### RT–qPCR

To verify the microarray results, RT–qPCR was performed on GV-stage oocytes from wild-type and knockout mice using the TaqMan® Gene Expression Cells-to-CT™ Kit (Ambion). Ten cells were washed

twice in PBS before lysis in 50 μl lysis/DNase buffer, giving a final lysate concentration of 0.2 cells/μl. 10 μl of cell lysate was added to 1× RT buffer for a final volume of 50 μl, resulting in a final concentration of 0.04 cell/μl. Gene expression was quantified using 4 μl of the resulting cDNA. Reactions were performed in 96-well plates on a StepOnePlus instrument (Applied Biosystems). Primers and probes (TaqMan Gene Expression assays) were ordered form Applied Biosystems and consist of a pair of unlabelled PCR primers and a TaqMan probe with a dye label (FAM) on the 5′ end, and a minor groove binder (MGB) and non-fluorescent quencher (NFQ) on the 3′ end. Normalisation was performed relative to *Gapdh* (fold-change found equal to 1 by microarray analysis). The $\Delta Ct$, which is determined by subtracting the Ct number for the reference gene from that of the target gene was statistically analysed. Relative quantification (RQ) was calculated ($2^{-\Delta\Delta Ct}$) and used to graphically present the results. Primers are listed in Appendix Table S1B.

## Phenotypic analysis of mutant mice

### Collection of GV and MII oocytes

GV oocytes were collected from 3- to 8-week-old females by puncturing ovaries with a 26-gauge needle in M2 medium (Sigma-Aldrich, Lyon, France) containing 150 μM dibutyryl cyclic AMP to prevent GV oocyte maturation and GVBD. Follicular cells surrounding GV oocytes were enzymatically (hyaluronidase 0.1%) and mechanically removed using a pipette with an inner diameter of around 100 μm. To extract GV oocytes from pre-antral follicles, follicles were treated with 2 mg/ml collagenase IV and 1 mg/ml hyaluronidase and ruptured with a fine pipette. For MII oocytes, 48 h after PMSG injection, 5 UI of human chorionic gonadotropin (hCG) was injected (Chorulon, Intervet), and cumulus-oocytes complexes (COCs) released from the ampullae were collected in M2 medium after 13 h. COCs were incubated in hyaluronidase enzyme (0.1 mg/ml, Sigma-Aldrich) for 5 min, and follicle-cell-free MII oocytes were obtained by pipetting.

### In vitro fertilisation (IVF)

Eggs were collected from 4- to 8-week-old females, synchronised by exposing to 5 units of PMSG (Synchropart, Intervet, Beaucouze, France) and 5 units of hCG. Sperm from healthy males (B6D2 F1) capacitated for 80 min in M16 + 2% BSA (A3803 Sigma-Aldrich) were simultaneously added to the COCs and incubated in M16 medium for 4 h at 37°C under 5% $CO_2$. Unbound sperm were washed away after 4-h incubation. 24 h after fertilisation, the different stages, that is unfertilised oocytes and two-cell embryos (as an indication of successful fertilisation), were scored. Embryos were cultured in potassium simplex optimised medium (KSOM) (1×, Life technologies) supplemented with essential and non-essential amino acids (KSOM/EAA): two-cell embryos were incubated in KSOM/EAA medium at 37°C under 5% $CO_2$ and cultured up to the blastocyst stage.

### Analysis of fertilised oocytes

After fertilisation, zygotes were transferred to a clean well containing M16 media and left for a further 2 h before being fixed in PFA 4% for 10 min and stained with Hoechst 33342 (2 μg/ml in PBS-PVP 0.5%). Oocytes were observed under confocal microscopy in drops of PBS-PVP covered with mineral oil on LabTeK chambered coverglass plates (Thermo Fisher Scientific, Villebon sur Yvette, France).

### Immunostaining of GV and MII oocytes

Cumulus-free MII or GV oocytes prepared as described above were fixed in 4% PFA for 15 min, washed twice in PBS-PVP (0.5%) and permeabilised in PBS-Triton 0.25% for 20 min. Oocytes were then blocked in blocking solution (2% NGS, 0.1% Triton) for 90 min at room temperature and stained overnight at 4°C with primary antibodies (Appendix Table S2). All antibodies were diluted in blocking solution. Following staining, oocytes were washed twice with PBS-PVP 0.5% and incubated for 1 h at 37°C in secondary antibody and Hoechst 33342 at 2 μl/ml in blocking solution. Oocytes were finally washed twice and observed by confocal microscopy (Zeiss LSM 710) in drops of PBS-PVP covered with mineral oil on LabTeK coverglass plates. Images were processed using Zen 2.1 software.

### Immunostaining of ovarian sections for oocyte counting

Mice were sacrificed by cervical dislocation, and ovaries were collected and fixed for 4 h in paraformaldehyde (4%). Ovaries were dehydrated embedded in paraffin, and 3-μm sections were cut. For histological analysis, after deparaffinisation, HIER and blocking slides were incubated with anti-Msy2 antibody (Appendix Table S2) for 1 h followed by a fluorophore-conjugated secondary antibody and Hoechst 33342 counterstaining. Stained sections were digitised at ×40 magnification using an Axioscope microscope (Zeiss, Germany) equipped with a motorised X–Y-sensitive stage. For each ovary, three sampling zones separated by 100–200 μm were studied. For each sampling zone, six to seven consecutive sections were stained and follicles counted, and the mean number of each class of follicle was calculated per sampling zone for WT and $Patl2^{-/-}$ mice.

## Protein extraction and Western Blot

WT and PATL2-HA-tagged mice were sacrificed by cervical dislocation, and the hypothalamus, pituitary glands and livers were collected, snap-frozen in liquid nitrogen and stored at −80°C. The day of the experiment, organs were thawed in RIPA lysis buffer (50 mM Tris–HCl pH 7.5, 150 mM NaCl, 1% NP-40, 0.5% sodium deoxycholate, Complete EDTA-free protease inhibitor cocktail tablet (Roche)) and homogenised using a Dounce tissue grinder before mixing the supernatant with loading buffer in equal volumes (4% SDS, 62.5 mM Tris pH 6.8, 0.1% bromophenol blue, 15% glycerol, 5% bromophenol blue). The cytoplasmic fraction was isolated by centrifugation at $3{,}000 \times g$ for 10 min. GV oocytes were collected in M2 medium supplemented with 150 mg/ml dbcAMP and prepared as described above. Oocytes were washed three times in PBS-PVP 0.5% to remove proteins from the M2 medium and added to an equal volume of loading buffer for a final volume of 20 μL. All protein samples were heated to 65°C for 15 min before loading onto a 4–20% TGX Mini-PROTEAN stain-free precast gel (Bio-Rad). Proteins were transferred onto a PVDF membrane, blocked in 5% milk and incubated overnight at 4°C in anti-HA antibody (Appendix Table S2) in blocking solution.

## Oocyte RNA profiling

Total RNAs were purified from wild-type or $Patl2^{-/-}$ mouse oocytes using Norgen Biotek's Single cell Purification kit (Cat. 51800; Thorold, ON, Canada) according to the manufacturer's protocol. The GV oocyte samples (two WT and three $Patl2^{-/-}$ samples)

## The paper explained

### Problem

Infertility is considered a global public health issue since it affects more than 70 million couples worldwide. The common treatment for infertile couples is IVF (*in vitro* fertilisation) or ICSI (intracytoplasmic sperm injection); however, both of these techniques require mature egg cells having correctly completed both meiotic divisions. In rare cases, women who undergo hormonal stimulation for IVF or ICSI produce only immature eggs which cannot be fertilised or made to mature *in vitro*. We named this condition "oocyte meiotic deficiency". This condition is very poorly understood.

### Results

We analysed a group of 23 women with oocyte meiotic deficiency by whole exome sequencing and identified a genetic mutation in the gene *PATL2* in 26% of patients. *PATL2* encodes an RNA-binding protein that has been shown to play an important role in egg cell maturation in frogs, but its function in mammals has not been studied until now. Using a knockout mouse model, we showed that Patl2 deficiency leads to defective oocyte maturation due to the deregulation of important genes involved in egg cell growth, meiosis and early embryo development. We also showed that Patl2 has a unique expression profile in comparison with other known oocyte RNA-binding proteins. These results indicate a specific, niche function for Patl2 in mRNA regulation during egg cell maturation in mammals.

### Impact

We have identified *PATL2* mutation as a major cause of oocyte meiotic deficiency. This finding can benefit patients through an improved understanding of their condition and allowing for better informed decisions regarding treatment options. We have also demonstrated that mouse Patl2 plays an important role in mRNA regulation, furthering our understanding of the process of mammalian oocyte maturation. This finding is highly relevant to the fast-growing field of clinical *in vitro* oocyte maturation, with wide-ranging applications including fertility treatment for patients with premature ovarian failure and fertility preservation for cancer patients facing reprotoxic chemotherapy.

contained a total number of cells ranging between 20–32 oocytes/sample; the MII oocyte samples (2 WT and 3 *Patl2*$^{-/-}$ samples) contained between 32 and 46 oocytes. Purified RNA concentrations were assessed by RT–qPCR using the Affymetrix RNA quantification kit. Sample RNAs (250 pg) were converted to biotin-labelled single-strand cDNAs using the Affymetrix Genechip WT pico kit. Labelled and fragmented ss-cDNA (5.5 µg) were hybridised to Affymetrix arrays (Genechip Clariom D mouse). The array complexity allowed analysis of the expression of more than 66,000 different transcripts (including transcript variants and non-coding-cRNA) by measuring the expression level of their individual exons.

The data obtained for the whole set of samples were normalised by applying the RMA process (Affymetrix Expression Consol). Raw and normalised data were uploaded to the GEO database (Accession number GSE100117). Probe-set annotation, quantitative expressions of all the transcripts and comparisons between the different groups of samples (GV-KO versus GV-WT; MII-KO versus MII-WT) were analysed using TAC.v3 software (Affymetrix). Expression levels are reported as log2 conversions of the intensities measured for the ss-cDNA hybridised to the arrays. Differential expression of transcripts between two groups of samples was considered significant when the expression ratios of the transcripts between the two groups were at least twofold higher or lower (in linear scale) with a *P*-value < 0.05 (ANOVA test).

### Transcriptomic analysis by IGA

IPA software (https://www.qiagenbioinformatics.com/products/ingenuity-pathway-analysis/) was used for the functional assessment of DET that were deregulated in *Patl2*$^{-/-}$ versus WT samples and to construct molecular interaction networks. IPA is software application that helps classify the molecular networks and functions most relevant to transcripts of interest. DET were imported to IPA for analysis. IPA generates pathways based on the transcripts contained in a data set and the information stored in the Ingenuity Pathways Knowledge Base. The significance of the transcripts' annotation is indicated by a *P*-value of < 0.05, as determined by a right-tailed Fisher's exact test.

### RT–qPCR validation experiments

Quantification of gene expression levels by RT–qPCR was performed on GV-stage oocytes from WT and *Patl2*$^{-/-}$ mice. cDNAs were produced from oocyte lysates using the TaqMan® Gene Expression Cells-to-CT™ Kit. Oocytes were isolated from the ovaries of 4-week-old mice (average of 50 GV per mouse) and washed twice in PBS-PVP 0.5% before lysis. Each lysis reaction was performed with 10 oocytes in a total volume of 50 µl of lysis buffer supplemented with DNase at a ratio of 1:100. RT reactions were performed using 10 µl of the lysate in a total volume of 50 µl using the same kit. Real-time PCR was performed in 96-well plates on the StepOnePlus system (Applied Biosystems). Reactions were performed in a total volume of 20 µl, comprising 4 µl of cDNA, 1 µl of TaqMan Assays and 10 µl of Master Mix. Data were normalised relative to expression levels measured for the *Gapdh* reference gene. ΔCt, which is determined by subtracting the Ct of the reference gene from that of the target gene, was the subject of statistical analysis.

### Statistical analyses

Statistical analyses were performed using SigmaPlot or GraphPad prism 6. Unpaired *t*-tests were used to compare WT and *Patl2*$^{-/-}$ samples. Data are represented as mean ± SEM. Statistical tests for which the two-tailed *P*-value ≤ 0.05 were considered significant.

### Data availability

Transcriptomic data are available in the GEO database: accession number GSE100117 (https://www.ncbi.nlm.nih.gov/geo/query/acc.cgi). Clinical exomic data are available in the EGA data base: EGAS00001002903 (https://www.ebi.ac.uk/ega/home).

**Expanded View** for this article is available online.

### Acknowledgements

We thank the IAB microscopy platform and Mylene PEZET, Alexei GRICHINE, Jacques MAZZEGA for their technical help. We thank Emeline FONTAINE PELLETIER (INSERM 1209, CNRS UMR 5309) for her generous donation of antibodies. We thank Marie-Christine BIRLING for helping with mouse genotyping. We thank Marcio CRUZEIRO (Institut Cochin, Paris, France) for providing the EIIaCre transgenic mice and Julien Fauré for access to molecular biology facility. Lastly, we thank all patients and control subjects for their participation.

We are also grateful to Jacques Brocard from the Photonic Imaging Centre at Grenoble Institute Neuroscience (Univ Grenoble Alpes—Inserm U1216—ISdV core facility and certified by the IBiSA label). This work was supported by the following grants: "Investigation of the genetic aetiology of oocyte meiotic failure (OMF) by exome sequencing" (awarded by the Fondation Maladies Rares (FMR) for the "High throughput sequencing and rare diseases 2012" programme), funding for the "MAS-Flagella" project (French National agency for research (ANR)) and the "Whole genome sequencing of patients with Flagellar Growth Defects (FGD)" (DGOS for the PRTS 2014 programme).

## Author contributions

CA and PFR analysed the data and wrote the manuscript; Z-EK, AA-Y, CC and MB performed molecular work; TK and NT-M analysed genetic data; MC-K performed IF and histological experiments; MC-K, JE, ELB and GM performed IVF experiments; JPI, AG and SA performed transcriptome analyses; EL and SPB performed biochemistry experiments; MC-K, EL and ELB performed histological study. SN, BC and Z-EK made HA-tagged mice. SFBM, IC-D, LH, OM, MM, HL, MK and RZ provided clinical samples and data; CA and PFR designed the study, supervised all laboratory work, had full access to all the data in the study and took responsibility for the integrity of the data and its accuracy. All authors contributed to the manuscript.

## Conflict of interest

The authors declare that they have no conflict of interest.

## For more information

OMIM gene number 614661: http://omim.org/entry/614661, phenotype entry 617743: http://omim.org/entry/617743.

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
