## [Review Process File · EMBO Molecular Medicine]

PATL2 is a key actor of oocyte maturation whose invalidation causes infertility in women and mice

Marie Christou-Kent, Zine-Eddine Kherraf, Amir Amiri-Yekta, Emilie Le Blévec, Thomas Karaouzène, Béatrice Conne, Jessica Escoffier, Said Assou, Audrey Guttin, Emeline Lambert, Guillaume Martinez, Magalie Boguenet, Selima Fourati Ben Mustapha, Isabelle Cedrin Durnerin, Lazhar Halouani, Ouafi Marrakchi, Mounir Makni, Habib Latrous, Mahmoud Kharouf, Charles Coutton, Nicolas Thierry-Mieg, Serge Nef, Serge P. Bottari, Raoudha Zouari, Jean Paul Issartel, Pierre F. Ray and Christophe Arnoult

Review timeline:	Submission date:	21 September 2017
	Editorial Decision:	2 November 2017
	Revision received:	14 February 2018
	Editorial Decision:	28 February 2018
	Revision received:	12 March 2018
	Accepted:	19 March 2018

Editor: Céline Carret

Transaction Report:

1st Editorial Decision

2 November 2017

Thank you for the submission of your manuscript to EMBO Molecular Medicine and please accept my apologies for the delay in getting back to you. We have been unusually busy and traveling lately, which, along with delays in securing referees reports and cross-commenting, has led to unfortunate delays. We have now heard back from the three referees whom we asked to evaluate your manuscript.

You will see that the three referees find the study to be of interest. However, novelty and limited oocyte maturation analyses are an issue. Furthermore, the study should benefit from rewriting to improve clarity and details (clinical and technological), but also structure. The recent publications in AJHG should be cited and discussed. Given the existing literature and data on *Xenopus* as pointed out by referee 3, the mice oocyte maturation defect should be analysed in the KO mice to improve novelty and provide functional insights. and ref. 2 and 3 provide suggestions for that. Please better describe the statistics and animal used as requested in our Author's checklist that you will have to fill.

Further, during our cross-commenting exercise, referee 3 suggested to strengthen the mouse data by comparing the microarray analysis to published microarray datasets from different stages of oocyte development, which in our view would be indeed informative and consolidate the findings.

We would therefore welcome the submission of a revised version within three months for further consideration and would like to encourage you to address all the criticisms raised as suggested to improve conclusiveness and clarity. Please note that EMBO Molecular Medicine strongly supports a single round of revision and that, as acceptance or rejection of the

manuscript will depend on another round of review, your responses should be as complete as possible.

I look forward to receiving your revised manuscript.

***** Reviewer's comments *****

Referee #1 (Comments on Novelty/Model System for Author):

The technical quality is high, using appropriate methodologies to address their questions. The novelty is medium to high as there are other groups addressing this (recent publication Chen et al) but this study is novel with respect to the patient group. Improvements would be if they could have compared results from North African population with another population but that would be difficult to achieve in this case.

Referee #1 (Remarks for Author):

This is an interesting study that deals with the clinical problem of oocyte meiotic deficiency which leads to infertility. The cause of this particular problem may be multi-factorial. This study carried out whole exome sequencing in 23 patients of North African origin and 6 patients had the same homozygous missense STOP mutation in PATL2.

The authors also carried out studies on Patl2 KO mice and demonstrated that this led to changes in expression of a number of genes associated with oocyte maturation and early embryonic development.

Another group previously identified that TUBB8 mutations were responsible for oocyte maturation arrest but that this only accounted for approx 30% of individuals with this phenotype. This would indicate that the causes of this condition are multi-factorial. In this study no deleterious TUBB8 variants were found therefore it is interesting to address whether the racial origin of the patient groups studied could account for the observed differences.

The group who observed TUBB8 as a factor have now published a follow up study that implicates Biallelic mutations in PAT12 in oocyte maturation arrest (Chen et al., 2017 Am J Hum Genet 101: 609-615). The Chen study and this current study both enhance our understanding of the genetic basis of this particular cause of female infertility.

The current study is a strong study particularly with the inclusion of the basic data from mouse, however, given the recently published work of Chen et al this paper should be re written to frame the Chinese results within the context of the results presented here. Particularly the differences in TUBB8 and the apparent differences in PATL2 mutations that affect granulosa cells (apparent in Chen et al paper but less so in this study).

Referee #2 (Comments on Novelty/Model System for Author):

In this study the authors have screened infertile women who show a failure in oocyte maturation. In a small cohort of predominantly Tunisian women, the authors identified a homozygous mutation in

PATL2. Furthermore, the authors subsequently show that *Patl2* deficient mice are subfertile, with reduced litter size and oocyte abnormalities. The authors also did expression profiling of oocytes of these *Patl2* deficient mice.

The paper is interesting as the identification of a novel gene involved in oocyte maturation and subsequent (partial) confirmation in a null mice. Hence this paper is definitely worth publishing. However, the paper requires substantial rewriting, in addition to some points that require clarification. The results section is lengthy, not very structured, and containing elements for the discussion section. Vice versa, the expression profiling results are described in the discussion section.

Referee #2 (Remarks for Author):

In this study the authors have identified a homozygous mutation in *PATL2* leading to a premature stop codon in women with a failure in oocyte maturation. The authors also show that in a mouse model, deficiency of this gene causes infertility due to oocyte abnormalities. The paper is interesting as the authors have identified a novel gene involved in oocyte maturation. There are, however, aspects of the paper that require further attention

- 1) The authors describe that 23 patients were analyzed. However, of these 23, 15 were analyzed by exome sequencing. Once the mutation was found, the remaining patients were sequenced for this mutations. The authors should explained why not all patients were screened by exome sequencing. Furthermore, this indicates that the screening for *TUBB8* was not performed in all patients.
- 2) In Table 1 the authors show the patient characteristics of the 6 patients in which the mutation was identified. The authors should show the characteristics of all patients in this table. Furthermore, it would be interesting to analyze whether differences exist in patients with or without this mutation, even though the cohort is small.
- 3) The authors may more clearly indicate that the patients were recruited in Tunisia (at least that is what I assume).
- 4) The authors used an in house bioinformatics pipeline. Please explain briefly or provided a reference to a previous paper in which this pipeline was used.
- 5) The authors should carefully indicate the number of mice used for all experiments. For some of the experiments the number is quite low, for instance in figure 3A, C and D
- 6) Figure 6 A and B. One of these figures is redundant and could be removed. If the authors would like to show the variation, they could plot the data as in figure 4B.
- 7) The authors should explain the targeting approach of the *Patl2* null mice. Which tissues does the *Ella-Cre* model target? Is *Patl2* also deleted in other tissues? This should be confirmed.
- 8) In the mouse model the authors only show a gross anatomy of the ovary. It would be useful to determine follicle number (of the different classes) and particularly number of Corpora Lutea in non-stimulated mice. Also information on gonadotropin levels would be useful.
- 9) The results section is quite long and contains elements that would be better suited for the discussion section. I would suggest to structure the results section more carefully. Furthermore, the results of the expression profiling is described mostly in the discussion section. This should be moved to the results section, with a proper pathway analysis.
- 10) For the expression profiling, the authors described several genes to be up- or down regulated. The authors should include validation studies for some of those genes using independent samples.

Referee #3 (Remarks for Author):

Here the authors use whole exome sequencing to identify a homozygous recessive mutation in the *PATL2* gene in 6 women from North Africa who presented to an assisted reproduction clinic with primary infertility, presumably out of a total of 23 patients who had a similar defect in generating

metaphase II stage oocytes during an assisted reproduction cycle. They go on to generate a full knockout for the homologous gene in mice. They show that the female mice are subfertile, likely because their oocytes are not fully grown and do not undergo normal meiotic maturation.

I have the following major concerns regarding this manuscript.

1. Although the final result regarding the human sequencing can be understood from the manuscript, the study was very poorly described. It is not clear how many patients were enrolled in the study (23 are stated but exome sequencing was done on only 15), whether they were enrolled before or after they had a diagnosis of failed oocyte maturation, and whether the IVF treatment cycle described in the Methods was performed before or after enrollment. It is never stated whether the IVF treatment cycle was part of the study or historical, or whether the oocytes were generated for research or clinical purposes. The 234 subjects who served as controls are never described, and there is no indication regarding whether or not they consented to the study. The diagnostic description for the 23 (or 15) subjects is incomplete - Table 1 only shows data regarding the 6 who had the PATL2 mutation. The entire description of the human aspects of this manuscript in the Methods and Results should be rewritten for clarity so the reader can understand what was done, and be reassured that the study was done following appropriate clinical research guidelines, particularly guidelines regarding informed consent.

Of note, a (very) recent publication in the American Journal of Human Genetics identified the same PATL2 mutation as in the current manuscript associated with female infertility due to oocyte maturation failure (Maddirevula et al., *Am J Hum Genet* 101:603, 2017; PMID 28965844). This paper only reported the human phenotype and did not pursue any mechanistic studies.

2. The evaluation of the *Padl2*^{-/-} mouse phenotype was extremely superficial and did not address the major underlying problem - lack of oocyte growth. It was demonstrated more than 25 years ago that oocytes that are not yet fully grown are not competent to undergo nuclear (meiotic) maturation and are not developmentally competent. Here the authors showed clearly that the oocytes in the female knockout mice were not fully grown. However, instead of pursuing studies to determine what was the function of the PADL2 protein during oocyte growth, the authors instead did studies that showed that the oocytes were not developmentally competent. At the end of the manuscript, the authors still have no idea what the protein does. Where the authors could have discovered information regarding oocyte growth was by doing a careful evaluation of ovarian morphology with time following birth. Instead, the authors stimulated the mice at 24 days of age with PMSG, which would impair an accurate evaluation of the phenotype at that age, and never examined ovaries of younger mice at all. A statement that the ovaries in knockout and WT mice had "no obvious defects" and had "similar cellular structures" is inadequate. There is a significant amount of published information regarding the function of PATL2 in *Xenopus* that could have been used to guide specific experiments to determine if the function was similar in the mouse. For example, was there any difference in the translation of any proteins essential for oocyte maturation? Does PATL2 interact with CPEB in the mouse?

3. The authors appear convinced that *Patl2* is an oocyte-specific transcript, but there is clearly a moderate level of expression in numerous additional tissues in both mouse and human (documented in Supplemental Figure S1). A function of PATL2 in other tissues could impact the fertility phenotype, and should be considered.

4. Statistical analysis - The analysis of the microarrays was apparently done by multiple T-tests as described in the Methods. Where was the correction for multiple comparisons? Was there an FDR cutoff? Given that only 2 WT samples were analyzed by microarray, how could a T-test be done?

Additional concerns:

1. English usage needs correction throughout the manuscript.
2. WES in the title should be spelled out, not abbreviated.
3. If the mouse genome has ~23,000 genes, how can the Affymetrix array detect 66,000 different genes? Perhaps what was meant was transcripts - based on alternate splicing?

You will see that the three referees find the study to be of interest. However, novelty and limited oocyte maturation analyses are an issue.

Furthermore, the study should benefit from rewriting to improve clarity and details (clinical and technological), but also structure.

Dear editor and reviewers.

Thank you for your constructive and encouraging feedback, which we have tried to incorporate into this new version of our manuscript.

I feel it is important to stress that our manuscript was submitted before the two articles reporting the effect of *PATL2* mutations on female reproduction were published in the American Journal of Human Genetics. We have been working on this project for over 4 years, and like the other two teams, we could have published our genetic data alone 2 years ago. However, we wanted to present a strong and detailed study including both human and mouse data. Because of this desire to present a comprehensive dataset (or as comprehensive as possible) we were unfortunately scooped. It is my understanding that EMBO journals' policy on scooping (that "similar findings published by other researchers during peer-review or revision of a manuscript are not used as a criterion for rejection") will not now prevent publication of our study. We hope that the reviewers and the editors will consider the information contained in our manuscript to be just as "novel" as that of our competitors.

We have now taken all of the reviewers very relevant comments into consideration and we feel that our manuscript is much improved (numerous new figures and data), which as highlighted by the reviewer contains considerably more work and information than the competing manuscripts. We hope that our article will now be accepted for prompt publication in Embo Mol Medicine.

The recent publications in AJHG should be cited and discussed.

The recent publications are now cited in the introduction section and their content was taken into account in the discussion

Given the existing literature and data on *Xenopus* as pointed out by referee 3, the mice oocyte maturation defect should be analysed in the KO mice to improve novelty and provide functional insights. and ref. 2 and 3 provide suggestions for that.

To respond to reviewers' queries, we have performed several new experiments since our initial study. These experiments were as follows:

- We were unable to find a specific mouse *Patl2* antibody, despite extensive efforts. To circumvent this problem, we used *CrsipRCas9* to generate a *Patl2*-HA tagged mice, using which we quantified the presence of *Patl2* in oocytes from primordial to MII stage. It is important to underline that tag/tag female animals are perfectly fertile, strongly suggesting that the insertion of HA tag in *Patl2* sequence does not alter its function
- We performed similar quantification experiment for different proteins known to be important for mRNA stability in oocytes, such as *Cpeb1*, *Msy2* and *Ddx6*.
- We performed experiments to determine whether these proteins co-localized with *Patl2*
- We measured the *Patl2* expression levels by Western Blot in the hypothalamic-pituitary axes.
- We performed a thorough analysis of the ovaries to determine the number of the different classes of oocytes at 12 and 26 days post-partum
- We analyzed the full cohort by WES sequencing

The results of these new experiments are described below.

Please better describe the statistics and animal used as requested in our Author's checklist that you will have to fill.

done

Further, during our cross-commenting exercise, referee 3 suggested to strengthen the mouse data by comparing the microarray analysis to published microarray datasets from different stages of oocyte development, which in our view would be indeed informative and consolidate the findings.

In our revised version, we have re-analyzed our transcriptomic data in the context of the GV-MII transition.

The results indicated that a large portion of DET in WT samples were involved in energy production (oxidative phosphorylation and mitochondrial (dys)function, Appendix figure S8) protein synthesis (EIF2 signaling, regulation of eIF4 and p70S6K signaling) and DNA replication, recombination and repair (Figure EV5-A).

The GV-MII transition was previously reported by Eppig's group as associated with specific degradation of a large number of transcripts associated with protein synthesis, DNA replication and energy production (Su et al, 2007). Reassuringly, we identified similar pathways as being regulated at this stage in our samples. The similarity of our results with those previously reported strengthens the quality of our raw data and validates our initial analysis in terms of the impact of the lack of Patl2 on the transcriptome. In addition to this transcriptome analysis, we performed RT-qPCR experiments for some up- and down-regulated transcripts to validate the transcriptomic results, as requested by reviewer 2. As a whole, these results rule out any bias in our transcriptomic analyses due to comparison of 2 WT samples with 3 KO samples at the GV stage.

Moreover, Figure EV5 presents a comparison of the pathways regulated in WT Patl2^{-/-} oocytes. Interestingly, the most significantly down- or up-regulated pathways were the same in WT and Patl2 KO mice, suggesting that Patl2 is not involved in the GV-MII transition.

We would therefore welcome the submission of a revised version within three months for further consideration and would like to encourage you to address all the criticisms raised as suggested to improve conclusiveness and clarity. Please note that EMBO Molecular Medicine strongly supports a single round of revision and that, as acceptance or rejection of the manuscript will depend on another round of review, your responses should be as complete as possible.

***** Reviewer's comments *****

Referee #1 (Comments on Novelty/Model System for Author):

The technical quality is high, using appropriate methodologies to address their questions. The novelty is medium to high as there are other groups addressing this (recent publication Chen et al) but this study is novel with respect to the patient group. Improvements would be if they could have compared results from North African population with another population but that would be difficult to achieve in this case.

Referee #1 (Remarks for Author):

This is an interesting study that deals with the clinical problem of oocyte meiotic deficiency which leads to infertility. The cause of this particular problem may be multi-factorial. This study carried out whole exome sequencing in 23 patients of North African origin and 6 patients had the same homozygous missense STOP mutation in PATL2.

The authors also carried out studies on Patl2 KO mice and demonstrated that this led to changes in expression of a number of genes associated with oocyte maturation and early embryonic development.

Another group previously identified that TUBB8 mutations were responsible for oocyte maturation arrest but that this only accounted for approx 30% of individuals with this phenotype. This would indicate that the causes of this condition are multi-factorial. In this study no deleterious TUBB8 variants were found therefore it is interesting to address whether the racial origin of the patient groups studied could account for the observed differences.

The group who observed TUBB8 as a factor have now published a follow up study that implicates Biallelic mutations in PATL2 in oocyte maturation arrest (Chen et al., 2017 Am J Hum Genet 101:

609-615). The Chen study and this current study both enhance our understanding of the genetic basis of this particular cause of female infertility.

The current study is a strong study particularly with the inclusion of the basic data from mouse, however, given the recently published work of Chen et al this paper should be re written to frame the Chinese results within the context of the results presented here.

The two articles presenting *PATL2* mutations as a cause of OMD are now presented in the introduction and their results are addressed in the discussion sections.

Particularly the differences in *TUBB8* and the apparent differences in *PATL2* mutations that affect granulosa cells (apparent in Chen et al paper but less so in this study).

In fact, the Chen paper clearly showed that *Patl2* is not expressed in granulosa cells (see figure S1). We obtained similar results: *Patl2* was undetectable by IF (*Patl2*-HA tag female) in the cumulus and is detectable only in oocytes.

Referee #2 (Comments on Novelty/Model System for Author):

In this study the authors have screened infertile women who show a failure in oocyte maturation. In a small cohort of predominantly Tunisian women, the authors identified a homozygous mutation in *PATL2*. Furthermore, the authors subsequently show that *Patl2* deficient mice are subfertile, with reduced litter size and oocyte abnormalities. The authors also did expression profiling of oocytes of these *Patl2* deficient mice.

The paper is interesting as the identification of a novel gene involved in oocyte maturation and subsequent (partial) confirmation in a null mice. Hence this paper is definitely worth publishing.

Thank you for this very positive overall assessment.

However, the paper requires substantial rewriting, in addition to some points that require clarification. The results section is lengthy, not very structured, and containing elements for the discussion section. Vice versa, the expression profiling results are described in the discussion section.

Referee #2 (Remarks for Author):

In this study the authors have identified a homozygous mutation in *PATL2* leading to a premature stop codon in women with a failure in oocyte maturation. The authors also show that in a mouse model, deficiency of this gene causes infertility due to oocyte abnormalities. The paper is interesting as the authors have identified a novel gene involved in oocyte maturation. There are, however, aspects of the paper that require further attention

- 1) The authors describe that 23 patients were analyzed. However, of these 23, 15 were analyzed by exome sequencing. Once the mutation was found, the remaining patients were sequenced for this mutations. The authors should explained why not all patients were screened by exome sequencing. Furthermore, this indicates that the screening for *TUBB8* was not performed in all patients.

Since our original submission we have completed the WES analysis for the remaining patients in our cohort to try to identify some new causal genes. This has been we clarified in the text. From this analysis we now have one or two new candidate genes of interest, and we plan to generate some KI mice to explore our new hypotheses. The characterization of these genes will take us at least two more years, so to limit the risks of being scooped (again) we prefer not to supply the list of genes identified. In the new version of our manuscript, we indicate that the genetic analysis was done in 3 steps (1, WES analysis of 15 patients; 2, *PATL2* Sanger analysis of the 8 additional subjects and 3, WES analysis of the remaining members of the cohort. However, we do describe the search for *TUBB8* mutations as we did identify one patient presenting a *TUBB8* loss of function heterozygote variant (in a *PATL2* non-mutated subject) during the second phase of WES analysis.

- 2) In Table 1 the authors show the patient characteristics of the 6 patients in which the mutation was identified. The authors should show the characteristics of all patients in this table. Furthermore, it would be interesting to analyze whether differences exist in patients with or without this mutation, even though the cohort is small.

Table 1 now presents the full cohort. We split the cohort into PATL2-dependent/independent oocyte maturation defects. Interestingly, PATL2 patients present a GV block, whereas patients with maturation defects not caused by PATL2 mutation tend rather to have a block at the MI stage (see new table 1).

- 3) The authors may more clearly indicate that the patients were recruited in Tunisia (at least that is what I assume).

The geographical origin of the patients is mainly Tunisia, but some patients are from Algeria, Libya, or France (but also of North-African descent). This is now indicated in the introduction and in table 1

- 3) The authors used an in house bioinformatics pipeline. Please explain briefly or provided a reference to a previous paper in which this pipeline was used.

We have added a reference to a publication where our bioinformatic pipeline is explained in detail (Coutton et al, Nature comm (2018)).

- 5) The authors should carefully indicate the number of mice used for all experiments. For some of the experiments the number is quite low, for instance in figure 3A, C and D

Unfortunately, we were unable to increase the numbers of mice used for figure 3A because it would take too long (9 months) to obtain and study new sexually-mature KO females (3 months to produce the animals, and 6 months to monitor fertility). Moreover, all the KO females available were used for other experiments requested by reviewers. However, we have included information on the numbers of animals used for each experiment.

- 6) Figure 6 A and B. One of these figures is redundant and could be removed. If the authors would like to show the variation, they could plot the data as in figure 4B.

Done

- 7) The authors should explain the targeting approach of the Patl2 null mice. Which tissues does the Ella-Cre model target?

Mice were provided by the EUCOM mutant mouse consortium, who generated numerous KO mice using the same strategy, consisting in inserting LacZ and neomycin cassettes in front of a crucial exon. The generators of this strain mice indicated that these cassette insertions should stop translation in 99% of cases. However, to be 100% sure that all mice would be Patl2 deficient, we chose to excise the cassettes in order to remove exon 7, thus producing a frameshift. Excision was performed by crossing Patl2 mice with the Ella-Cre model. This mouse line carries a Cre transgene under the control of the promoter for the EIIa adenovirus which targets expression of Cre recombinase to early mouse embryos. With this construction, Cre expression is thought to occur prior to implantation in the uterine wall. Cre-mediated recombination occurs in a wide range of tissues, including the germ cells that transmit the genetic alteration to offspring. The presence of a frameshift was validated by PCR and sequencing in F0 and F1 mice. Experiments were performed using F1-F3 animals to rule out the possibility of mosaicism. This information is now included in the methods section.

Is Patl2 also deleted in other tissues? This should be confirmed.

Several generations of Patl2 KO mice were studied, and because this mouse line was bred as heterozygous animals, all animals were selected through genotyping performed on skin, muscle and bones tissues (fingertip). The deletion is therefore not restricted to the ovaries, but affects all tissues.

This question intersects with issue 3 raised by referee 3, suggesting that reproductive effect of the lack of Patl2 could be due to expression of Patl2 in other tissues. As explained below, we think this is unlikely.

8) In the mouse model the authors only show a gross anatomy of the ovary. It would be useful to determine follicle number (of the different classes) and particularly number of Corpora Lutea in non-stimulated mice.

This is a very important remark and we agree that our study was too preliminary to draw a robust conclusion.

In the revised version, we provide a thorough analysis of the comparative number of different types of follicles identified at 12 and 26 days post-partum from non-stimulated females. All ovaries were sectioned and we counted the mean number of follicles (primordial, primary, secondary follicles) found in a 3 μ m thick sample. Between 40 and 50 sections were counted per genotype at 12 and 26 dpp.

Also information on gonadotropin levels would be useful.

In women, as shown in table 1, the level of LH, FSH and TSH were normal, suggesting that the hypothalamic–pituitary axes and in particular the gonadal axis are not affected by the Patl2 mutation. Moreover, the women harboring the Patl2 mutation presented a normal menstrual cycle. Taken together, these results strongly suggest that hormonal control of reproduction is unaffected in these patients.

To validate this conclusion, we now present comparative Western-Blot analysis of HA-Tagged-Patl2 mice performed with extracts from oocytes, the hypothalamus and pituitary gland. Whereas a clear and specific signal is observed in the oocyte extract, no bands were observed in the hypothalamus or pituitary gland extracts, confirming that Patl2 does not modulate the hypothalamic–pituitary axes. These data are now presented in supplementary figure S2 and this result is discussed in the manuscript.

9) The results section is quite long and contains elements that would be better suited for the discussion section. I would suggest to structure the results section more carefully. Furthermore, the results of the expression profiling is described mostly in the discussion section. This should be moved to the results section, with a proper pathway analysis.

We have completely rewritten this part of the paper, and we now include a pathways analysis (Figure EV4)

10) For the expression profiling, the authors described several genes to be up- or down regulated. The authors should include validation studies for some of those genes using independent samples.

We used RT-qPCR to test the expression levels for very meaningful genes identified as down- or up-regulated in the transcriptomic studies. These new data confirmed the validity of the transcriptomic analysis. These data were obtained from independent WT and KO mice and are presented in supplementary figure S7.

Referee #3 (Remarks for Author):

Here the authors use whole exome sequencing to identify a homozygous recessive mutation in the PATL2 gene in 6 women from North Africa who presented to an assisted reproduction clinic with primary infertility, presumably out of a total of 23 patients who had a similar defect in generating metaphase II stage oocytes during an assisted reproduction cycle. They go on to generate a full knockout for the homologous gene in mice. They show that the female mice are subfertile, likely because their oocytes are not fully grown and do not undergo normal meiotic maturation.

I have the following major concerns regarding this manuscript.

1. Although the final result regarding the human sequencing can be understood from the manuscript, the study was very poorly described. It is not clear how many patients were enrolled in the study (23 are stated but exome sequencing was done on only 15), whether they were enrolled before or after they had a diagnosis of failed oocyte maturation, and whether the IVF treatment cycle described in the Methods was performed before or after enrollment. It is never stated whether the IVF treatment cycle was part of the study or historical, or whether the oocytes were generated for research or clinical purposes. The 234 subjects who served as controls are never described, and there is no indication regarding whether or not they consented to the study. The entire description of the human aspects of this manuscript in the Methods and Results should be rewritten for clarity so the reader can understand what was done, and be reassured that the study was done following appropriate clinical research guidelines, particularly guidelines regarding informed consent.

For this study, patients received no specific treatment for research purposes, and all oocytes collected were used solely for IVF procedures. After IVF treatment, performed independently of the research, couples of interest were selected and referred to us by their physician. After the physician had explained the study and its aims, subjects signed an informed consent document. DNA samples were collected from the couples of interests, and the patients' identity was coded so that subjects could not be identified.

Since 2007 and in parallel to this study on female infertility, we also performed a large study on male infertility, involving several hundred couples (approximately 400 patients have already been sequenced) and using the same ethical protocol. Most of these couples had IVF-ICSI at the Clinic des Jasmin in Tunis from which 21/23 of the patients described here were recruited and had IVF. Female partners of infertile men, for whom no fertility problems were identified were considered fertile controls, and their characteristics were recorded and used anonymously. The male infertility cohort has already been partly presented in several papers, and all projects were authorized by the ethics committee at Grenoble Hospital

- Dieterich et al (2007) Homozygous mutation of *AURKC* yields large-headed polyploid spermatozoa and causes male infertility. *Nat Genet.* 2007 May;39(5):661-5
- Harbuz et al (2011) A recurrent deletion of the *DPY19L2* gene causes infertility in man by blocking sperm head elongation and acrosome formation. *Am. J. Hum. Genet.* 88: 351-361
- Ben Khelifa et al (2014) Mutations in *DNAH1*, which encodes an inner arm heavy chain dynein, lead to male infertility from multiple morphological abnormalities of the sperm flagella *Am. J. Hum. Genet.* 94: 95-104.
- Coutton et al Ray (2018) Mutations in CFAP43 and CFAP44 cause male infertility and flagellum defects in Trypanosoma and human. *Nature Comm*, DOI: 10.1038/s41467-017-02792-7

These points are now clearly mentioned in the methods section.

The diagnostic description for the 23 (or 15) subjects is incomplete - Table 1 only shows data regarding the 6 who had the PATL2 mutation.

The table now contains information for the full cohort, and all patients have been sequenced.

Of note, a (very) recent publication in the American Journal of Human Genetics identified the same PATL2 mutation as in the current manuscript associated with female infertility due to oocyte maturation failure (Maddirevula et al., *Am J Hum Genet* 101:603, 2017; PMID 28965844). This paper only reported the human phenotype and did not pursue any mechanistic studies.

Thank you for pointing this paper out to us. Although we are naturally disappointed to have been scooped by this articles and the one by Chen et al., both of which were published after submission of our manuscript, we agree that our study is more complete than these other studies. The data we presented for the KO mouse remains novel, and our analysis is more complete, as indicated by the reviewer. Most journals, and EMBO journals in particular, have a clear policy on scooping, indicating that "similar findings published by other researchers during peer-review or revision of a manuscript are not used as a criterion for rejection". In the revised version of our manuscript, we now cite the two AJHG articles in the introduction and discussion.

2. The evaluation of the *Patl2*^{-/-} mouse phenotype was extremely superficial and did not address the major underlying problem - lack of oocyte growth. It was demonstrated more than 25 years ago that oocytes that are not yet fully grown are not competent to undergo nuclear (meiotic) maturation and are not developmentally competent. Here the authors showed clearly that the oocytes in the female knockout mice were not fully grown. However, instead of pursuing studies to determine what was the function of the PATL2 protein during oocyte growth, the authors instead did studies that showed that the oocytes were not developmentally competent. At the end of the manuscript, the authors still have no idea what the protein does. Where the authors could have discovered information regarding oocyte growth was by doing a careful evaluation of ovarian morphology with time following birth. Instead, the authors stimulated the mice at 24 days of age with PMSG, which would impair an accurate evaluation of the phenotype at that age, and never examined ovaries of younger mice at all. A statement that the ovaries in knockout and WT mice had "no obvious defects" and had "similar cellular structures" is inadequate

We now provide a thorough analysis of the comparative number of the different types of follicles present in ovaries at 12 and 26 days postpartum from non-stimulated females (Figure EV2 and supplementary figure S4). All ovaries were sectioned and we counted the mean number of follicles (primordial, primary, secondary) found in 3 μ m thick samples. The different classes of follicle oocytes (primordial, primary and secondary) were counted in 9 different 3- μ m sections from 3 different mice (for each section, 4-7 technical replicates, corresponding to successive sections were counted).

To better characterize the function of *Patl2* in mouse oocytes, we studied its localization, its presence and its abundance in the different types of growing, GV and MII oocytes using IF and confocal microscopy to measure the intensity of fluorescence for the different oocyte types from HA-tagged *Patl2* mice. Figure 2 shows the results obtained. The specificity of the fluorescent signal was validated by measuring fluorescence in WT oocytes, where no signal was measured (Appendix Figure S5).

There is a significant amount of published information regarding the function of PATL2 in *Xenopus* that could have been used to guide specific experiments to determine if the function was similar in the mouse. For example, was there any difference in the translation of any proteins essential for oocyte maturation? Does PATL2 interact with CPEB in the mouse?

First, we quantified the presence/abundance of different proteins known to be involved in mRNA stability such as *Cpeb1*, *Msy2* and *Ddx6* in the different type of oocytes. The results revealed no overlap in staining pattern with *Patl2*: *Patl2* was not detectable in primordial oocytes and its abundance was higher in secondary preantral oocytes.

In addition, colocalization experiments with these proteins revealed no colocalization with *Patl2* in mouse oocytes. The results of these experiments are now presented in figures 2 and 3. It is worth noting that HA is a small tag (9 amino acids) compared to GFP and is therefore less likely to induce tag-dependent relocalization. Moreover, normal fertility in natural mating was recorded for homozygous *Patl2*-HA females.

These results suggest that *Patl2* represents an alternative mRNA stability pathway in mammalian oocytes.

3. The authors appear convinced that *Patl2* is an oocyte-specific transcript, but there is clearly a moderate level of expression in numerous additional tissues in both mouse and human (documented in Supplemental Figure S1). A function of PATL2 in other tissues could impact the fertility phenotype, and should be considered.

As stated above, in women, as shown in table 1, the level of LH, FSH and TSH were normal, suggesting that the hypothalamic-pituitary axes and in particular the gonadal axis were unaffected by the absence of *Patl2*. Moreover, no alterations to menstrual cycles were observed in patients harboring the PATL2 mutation. Altogether, these results strongly suggest that the hormonal control of reproduction is not affected in these patients.

To validate this conclusion, we now present comparative Western-Blot analysis of HA-Tagged-*Patl2* mice performed with Oocytes, hypothalamus and pituitary gland extracts. Whereas a clear and

specific signal is observed with oocyte extracts, no bands were observed with hypothalamus and pituitary gland extracts, confirming that Patl2 does not modulate the hypothalamic–pituitary axes. These data are now presented in supplementary figure 3 and this point is now discussed in the discussion section.

4. Statistical analysis - The analysis of the microarrays was apparently done by multiple T-tests as described in the Methods. Where was the correction for multiple comparisons? Was there an FDR cutoff?

TAC performs statistical analysis to obtain a list of differentially expressed genes. It also provides several graphical outputs for the visualization and appraisal of the mRNA deregulation events. TAC runs analyses based on Expression Console (EC)-generated CHP files. Multi-testing correction is performed using the Benjamini-Hochberg Step-Up FDR-controlling procedure for all the genes expressed. By default, the Alpha level is set to 0.05 in the TAC Parameters (False Discovery Rate field) or the FDR-adjusted p-value is set to 0.05

Given that only 2 WT samples were analyzed by microarray, how could a T-test be done?

As explained by statisticians from the GraphPad project (<https://www.graphpad.com/support/faqid/591/>), the equations used to calculate the SD, SEM and CI all work fine when you have only duplicate (N=2) data. It is therefore valid to compute a t-test with only two replicates in each group, although the statistical power is greater with a higher number of samples.

Moreover, we now have several corroborating results confirming that the genes identified as down/up-regulated at the GV stage are not stochastic results

- RT-qPCR experiments confirmed the results for all genes tested
- Most of the meaningful genes found to be downregulated at the GV stage were also down regulated at the MII stage (with n=3 per sample)
- GV-MII transition in WT and KO oocytes gave the same down and up regulated pathways

Additional concerns:

1. English usage needs correction throughout the manuscript.

Done

2. WES in the title should be spelled out, not abbreviated.

The title was changed to respect size limitation. WES is no longer used

3. If the mouse genome has ~23,000 genes, how can the Affymetrix array detect 66,000 different genes? Perhaps what was meant was transcripts - based on alternate splicing?

This was corrected, the Affymetrix array detects 66,000 transcripts.

2nd Editorial Decision

28 February 2018

Thank you for the submission of your revised manuscript to EMBO Molecular Medicine. We have now received the enclosed reports from the referees that were asked to re-assess it. As you will see the reviewers are now globally supportive and I am pleased to inform you that we will be able to accept your manuscript pending minor editorial changes, and a response to the referee.

Please address the minor comment of referee 2 in writing. Please provide a letter INCLUDING my comments and the reviewer's reports and your detailed responses to their comments (as Word file).

Please submit your revised manuscript within two weeks. I look forward to seeing a revised form of your manuscript as soon as possible.

***** Reviewer's comments *****

Referee #2 (Comments on Novelty/Model System for Author):

The authors have identified a mutation in PatL2 in infertile patients. This relatively novel mutation was subsequently analyzed in a knockout mouse model. This data provide insight into the biological stage in oocyte maturation affected by absence of PatL2 function.

Referee #2 (Remarks for Author):

The authors have put a significant effort into improving their manuscript. The data are now presented in a logical and clear way, although the manuscript remains quite lengthy. Most of my comments have been addressed. Some comments remain.

1. In my previous comment 8 I asked for information on gonadotropin levels in the knockout mice. The authors reasoned that since gonadotropin levels in patients and PatL2 is not expressed in the hypothalamus and pituitary of mice, it is not expected that gonadotropin levels are affected. Given that species differences exist, direct comparison of human and mouse data may not be valid. Furthermore, the number of human samples for which gonadotropin data are presented is very limited. I would have expected that since these patients underwent IVF, this data would be available. Furthermore, lack of PatL2 expression in the brain, doesn't exclude an effect of the ovary on the HPG-axis. A defect in oocyte-granulosa cell communication might results in altered granulosa cell functioning for instance, which subsequently could affect the negative feedback. The authors should be less strong in their conclusion.

2. In figure 8, the authors have drawn dotted lines between WT and PatL2-KO mice. Please remove these lines as these are independent mice, and not mice that have been studied at 2 time points.

Referee #3 (Comments on Novelty/Model System for Author):

Experiments are now well described and performed appropriately and thoroughly, new and interesting data are reported regarding the mechanisms underlying a clinically relevant phenotype in humans based on excellent work in a mouse model, and the information gleaned is highly relevant for clinical assisted reproduction.

Referee #3 (Remarks for Author):

In the revised manuscript, the authors have done an outstanding job of delving into the mechanisms underlying the Patl2^{-/-} phenotype in the mouse. The new experiments added, along with a thorough discussion of the implications of the results, dramatically improved the impact of the manuscript and strongly differentiate this manuscript from two previously published papers that simply reported a human Patl2 mutation in association with a clinical infertility phenotype.

My concerns regarding the description of the procedures and consent process for the human subjects, possible influence of Patl2 knockout on somatic tissues, and statistical analysis were all satisfactorily addressed.

2nd Revision - authors' response

12 March 2018

We are delighted with the decision to accept our manuscript for publication in EMBO Molecular Medicine. We have made all the suggested final amendments and hope to have satisfactorily addressed all comments.

Our detailed responses :

1) Please address the minor comment of referee 2 in writing. Please provide a letter INCLUDING my comments and the reviewer's reports and your detailed responses to their comments (as Word file).

Done

***** Reviewer's comments *****

Referee #2 (Comments on Novelty/Model System for Author):

The authors have identified a mutation in PatL2 in infertile patients. This relatively novel mutation was subsequently analyzed in a knockout mouse model. This data provide insight into the biological stage in oocyte maturation affected by absence of PatL2 function.

Referee #2 (Remarks for Author):

The authors have put a significant effort into improving their manuscript. The data are now presented in a logical and clear way, although the manuscript remains quite lengthy. Most of my comments have been addressed. Some comments remain.

1. In my previous comment 8 I asked for information on gonadotropin levels in the knockout mice. The authors reasoned that since gonadotropin levels in patients and PatL2 is not expressed in the hypothalamus and pituitary of mice, it is not expected that gonadotropin levels are affected. Given that species differences exist, direct comparison of human and mouse data may not be valid. Furthermore, the number of human samples for which gonadotropin data are presented is very limited. I would have expected that since these patients underwent IVF, this data would be available. Furthermore, lack of PatL2 expression in the brain, doesn't exclude an effect of the ovary on the HPG-axis. A defect in oocyte-granulosa cell communication might results in altered granulosa cell functioning for instance, which subsequently could affect the negative feedback. The authors should be less strong in their conclusion.

To take into account the reviewer's comment, we have modified the corresponding text, which is now : Whereas a clear and specific signal is observed for oocyte extracts from PATL2-HA females, no signal was observed in extracts from the hypothalamus or pituitary gland indicating that the direct control of the hypothalamus/pituitary gland on oocyte maturation is not altered in mice (Appendix Figure S2). Since our PATL2 patients exhibited normal hormone levels (when data were available, Table 1) and reported regular menstrual cycles, these results taken together suggest that the human infertility phenotype is purely due to an oocyte defect.

2. In figure 8, the authors have drawn dotted lines between WT and PatL2-KO mice. Please remove these lines as these are independent mice, and not mice that have been studied at 2 time points.

We have changed the format of this graph to avoid confusion (assuming the figure in question is in fact figure 6A for which this comment would be appropriate).

Referee #3 (Comments on Novelty/Model System for Author):

Experiments are now well described and performed appropriately and thoroughly, new and interesting data are reported regarding the mechanisms underlying a clinically relevant phenotype in humans based on excellent work in a mouse model, and the information gleaned is highly relevant for clinical assisted reproduction.

Referee #3 (Remarks for Author):

In the revised manuscript, the authors have done an outstanding job of delving into the mechanisms underlying the Patl2^{-/-} phenotype in the mouse. The new experiments added, along with a thorough discussion of the implications of the results, dramatically improved the impact of the manuscript and strongly differentiate this manuscript from two previously published papers that simply reported a human Patl2 mutation in association with a clinical infertility phenotype.

My concerns regarding the description of the procedures and consent process for the human subjects, possible influence of Patl2 knockout on somatic tissues, and statistical analysis were all satisfactorily addressed.

Thank you for this evaluation.

Corresponding Author Name: Christophe ARNOULT

Journal Submitted to: Embo Mol Med

Manuscript Number: EMM-2017-08515